# IMPROVED DETERMINISTIC $l_2$ ROBUSTNESS ON CIFAR-10 AND CIFAR-100

**Sahil Singla**[1]**, Surbhi Singla**[2]**, Soheil Feizi**[1]
University of Maryland, College Park
`{ssingla,sfeizi}@umd.edu`[1], `surbhisingla1995@gmail.com`[2]

## ABSTRACT

Training convolutional neural networks (CNNs) with a strict Lipschitz constraint under the $l_2$ norm is useful for provable adversarial robustness, interpretable gradients and stable training. While 1-Lipschitz CNNs can be designed by enforcing a 1-Lipschitz constraint on each layer, training such networks requires each layer to have an orthogonal Jacobian matrix (for all inputs) to prevent the gradients from vanishing during backpropagation. A layer with this property is said to be Gradient Norm Preserving (GNP). In this work, we introduce a procedure to certify the robustness of 1-Lipschitz CNNs by relaxing the orthogonalization of the last linear layer of the network that significantly advances the state of the art for both standard and provable robust accuracies on CIFAR-100 (gains of $4.80\%$ and $4.71\%$, respectively). We further boost their robustness by introducing (i) a novel Gradient Norm preserving activation function called the Householder activation function (that includes every GroupSort activation) and (ii) a certificate regularization. On CIFAR-10, we achieve significant improvements over prior works in provable robust accuracy ($5.81\%$) with only a minor drop in standard accuracy ($-0.29\%$). Code for reproducing all experiments in the paper is available at `https://github.com/singlasahil14/SOC`.

## 1 INTRODUCTION

Given a neural network $f : \mathbb{R}^d \to \mathbb{R}^k$, the Lipschitz constant[1] $\mathrm{Lip}(f)$ enforces an upper bound on how much the output is allowed to change in proportion to a change in the input. Previous work has demonstrated that a small Lipschitz constant is useful for improved adversarial robustness (Szegedy et al., 2014; Cissé et al., 2017), generalization bounds (Bartlett et al., 2017; Long & Sedghi, 2020), interpretable gradients (Tsipras et al., 2018) and Wasserstein distance estimation (Villani, 2008). $\mathrm{Lip}(f)$ also upper bounds the increase in the norm of gradient during backpropagation and can thus prevent gradient explosion during training, enabling us to train very deep networks (Xiao et al., 2018). While heuristic methods to enforce Lipschitz constraints (Miyato et al., 2018; Gulrajani et al., 2017) have achieved much practical success, they do not provably enforce a bound on $\mathrm{Lip}(f)$ globally and it remains challenging to achieve similar results when $\mathrm{Lip}(f)$ is provably bounded.

Using the property: $\mathrm{Lip}(g \circ h) \leq \mathrm{Lip}(g)\,\mathrm{Lip}(h)$, the Lipschitz constant of the neural network can be bounded by the product of the Lipschitz constant of all layers. While this allows us to construct 1-Lipschitz neural networks by constraining each layer to be 1-Lipschitz, Anil et al. (2018) identified a key difficulty with this approach. Because a 1-Lipschitz layer can only reduce the norm of gradient during backpropagation, backprop through each layer reduces the gradient norm, resulting in small gradient values for layers closer to the input, making training slow and difficult. To address this problem, they introduce Gradient Norm Preserving (GNP) architectures where each layer preserves the gradient norm during backpropagation. This involves constraining the Jacobian of each linear layer to be an orthogonal matrix and using a GNP activation function called GroupSort. GroupSort activation function (Anil et al., 2018) first separates the vector of preactivations $\mathbf{z} \in \mathbb{R}^m$ into groups of pre-specified sizes, sorts each group in the descending order and then concatenates these sorted groups. When the group size is 2, the resulting activation function is called MaxMin.

---

[1]Unless specified, we assume the Lipschitz constant under the $l_2$ norm in this work.

For 1-Lipschitz CNNs, the robustness certificate for a sample $\mathbf{x}$ from class $l$ is computed as $\mathcal{M}_f(\mathbf{x})/\sqrt{2}$ where $\mathcal{M}_f(\mathbf{x}) = f_l(\mathbf{x}) - \max_{i \neq l} f_i(\mathbf{x})$. Naturally, larger values of $f_l(\mathbf{x})$ and smaller values of $\max_{j \neq l} f_j(\mathbf{x})$ will lead to larger certificates. This requires the last weight matrix in the network, denoted by $\mathbf{W} \in \mathbb{R}^{k \times m}$ ($k$ is the number of classes, $m$ is the dimension of the penultimate layer, $m > k$), to enforce the following constraints throughout training:

$$\forall j, \ \|\mathbf{W}_{j,:}\|_2 = 1, \qquad i \neq j, \ \mathbf{W}_{j,:} \perp \mathbf{W}_{i,:}$$

where $\mathbf{W}_{i,:}$ denotes the $i^{th}$ row of $\mathbf{W}$. Now suppose that for some input image with label $l$, we want to update $\mathbf{W}$ to increase the logit for the $l^{th}$ class. Since $\|\mathbf{W}_{l,:}\|_2$ is constrained to be 1, the logit can only be increased by changing the direction of the vector $\mathbf{W}_{l,:}$. Because the other rows $\{\mathbf{W}_{i,:}, \ i \neq l\}$ are constrained to be orthogonal to $\mathbf{W}_{l,:}$, this further requires an update for all the rows of $\mathbf{W}$. Thus during training, any update made to learn some class must necessitate the forgetting of information relevant for the other classes. This can be particularly problematic when the number of classes $k$ and thus the number of orthogonality constraints per row (i.e., $k - 1$) is large (such as in CIFAR-100).

To address this limitation, we propose to keep the last weight layer of the network *unchanged*. But then the resulting function is no longer 1-Lipschitz and the certificate $\mathcal{M}_f(\mathbf{x})/\sqrt{2}$ is not valid. Thus, we introduce a new certification procedure that does not require the last weight layer of the network $\mathbf{W}$ to be orthogonal. Our certificate is then given by the following equation:

$$\min_{i \neq l} \frac{f_l(\mathbf{x}) - f_i(\mathbf{x})}{\|\mathbf{W}_{l,:} - \mathbf{W}_{i,:}\|_2}$$

However, a limitation of using the above certificate is that because the weight layers are completely unconstrained, larger norms of rows (i.e., $\|\mathbf{W}_{i,:}\|$) can result in larger values of $\|\mathbf{W}_{l,:} - \mathbf{W}_{i,:}\|_2$ and thus smaller certificate values. To address this limitation, we normalize all rows to be of unit norm before computing the logits. While this still requires all the rows to be of unit norm, their directions are now allowed to change freely thus preventing the need to update other rows and forgetting of learned information. We show that this provides significant improvements when the number of classes is large. We call this procedure *Last Layer Normalization* (abbreviated as *LLN*). On CIFAR-100, this significantly improves both the standard ($> 3\%$) and provable robust accuracy ($> 4\%$ at $\rho = 36/255$) across multiple 1-Lipschitz CNN architectures (Table 1). Here, $\rho$ is the $l_2$ attack radius.

Another limitation of existing 1-Lipschitz CNNs (Li et al., 2019b; Trockman & Kolter, 2021; Singla & Feizi, 2021) is that their robustness guarantees do not scale properly with the $l_2$ radius $\rho$. For example, the provable robust accuracy of (Singla & Feizi, 2021) drops $\sim 30\%$ at $\rho = 108/255$ compared to $36/255$ on CIFAR-10 (Table 2). To address this limitation, we introduce a certificate regularization denoted by CR (Section 5) that when used along with the Householder activation results in significantly improved provable robust accuracy at larger radius values with minimal loss in standard accuracy. On the CIFAR-10 dataset, we achieve significant improvements in the provable robust accuracy for large $\rho = 108/255$ (min gain of $+4.96\%$) across different architectures with minimal loss in the standard accuracy (max drop of $-0.56\%$). Results are in Table 2.

Additionally, we characterize the $\mathrm{MaxMin}$ activation function as a special case of the more general *Householder* (HH) activations. Recall that given $\mathbf{z} \in \mathbb{R}^m$, the HH transformation is a linear function reflecting $\mathbf{z}$ about the hyperplane $\mathbf{v}^T\mathbf{x} = 0$ ($\|\mathbf{v}\|_2 = 1$), given by $(\mathbf{I} - 2\mathbf{v}\mathbf{v}^T)\mathbf{z}$ where $\mathbf{I} - 2\mathbf{v}\mathbf{v}^T$ is orthogonal because $\|\mathbf{v}\|_2 = 1$. The *Householder* activation function $\sigma_{\mathbf{v}}$ is defined below:

$$\sigma_{\mathbf{v}}(\mathbf{z}) = \begin{cases} \mathbf{z}, & \mathbf{v}^T\mathbf{z} > 0, \\ (\mathbf{I} - 2\mathbf{v}\mathbf{v}^T)\mathbf{z}, & \mathbf{v}^T\mathbf{z} \leq 0. \end{cases} \tag{1}$$

First, note that since $\mathbf{z} = (\mathbf{I} - 2\mathbf{v}\mathbf{v}^T)\mathbf{z}$ along $\mathbf{v}^T\mathbf{z} = 0$, $\sigma_{\mathbf{v}}$ is continuous. Moreover, the Jacobian $\nabla_{\mathbf{z}} \sigma_{\mathbf{v}}$ is either $\mathbf{I}$ or $\mathbf{I} - 2\mathbf{v}\mathbf{v}^T$ (both orthogonal) implying $\sigma_{\mathbf{v}}$ is GNP. Since these properties hold $\forall \, \mathbf{v} : \|\mathbf{v}\|_2 = 1$, $\mathbf{v}$ can be learned during the training. In fact, we prove that any GNP piecewise linear function that changes from $\mathbf{Q}_1\mathbf{z}$ to $\mathbf{Q}_2\mathbf{z}$ ($\mathbf{Q}_1, \mathbf{Q}_2$ are square orthogonal matrices) along $\mathbf{v}^T\mathbf{z} = 0$ must satisfy $\mathbf{Q}_2 = \mathbf{Q}_1 (\mathbf{I} - 2\mathbf{v}\mathbf{v}^T)$ to be continuous (Theorem 1). Thus, this characterization proves that every $\mathrm{GroupSort}$ activation is a special case of the more general Householder activation function (example in Figure 1, discussion in Section 6).

In summary, in this paper, we make the following contributions:

- We introduce a certification procedure *without* orthogonalizing the last linear layer called *Last Layer Normalization*. This procedure significantly enhances the standard and provable robust accuracy when the number of classes is large. Using the LipConvnet-15 network on CIFAR-100, our modification achieves a gain of $+4.71\%$ in provable robust accuracy (at $\rho = 36/255$) with a gain of $+4.80\%$ in standard accuracy (Table 1).

- We introduce a *Certificate Regularizer* that significantly advances the provable robust accuracy with a small reduction in standard accuracy. Using LipConvnet-15 network on CIFAR-10, we achieve $+5.81\%$ improvement in provable robust accuracy (at $\rho = 108/255$) with only a $-0.29\%$ drop in standard accuracy over the existing methods (Table 2).

- We introduce a class of piecewise linear GNP activation functions called *Householder* or HH activations. We show that the $\mathrm{MaxMin}$ activation is a special case of the HH activation for certain settings. We prove that Householder transformations are *necessary* for any GNP piecewise linear function to be continuous (Theorem 1).

## 2 RELATED WORK

**Provably Lipschitz convolutional neural networks**: The class of fully connected neural networks (FCNs) which are Gradient Norm Preserving (GNP) and 1-Lipschitz were first introduced by Anil et al. (2018). They orthogonalize weight matrices and use $\mathrm{GroupSort}$ as the activation function to design each layer to be GNP. While there have been numerous works on enforcing Lipschitz constraints on convolution layers (Cissé et al., 2017; Tsuzuku et al., 2018; Qian & Wegman, 2019; Gouk et al., 2020; Sedghi et al., 2019), they either enforce loose Lipschitz bounds or are not scalable to large networks. To ensure that the Lipschitz constraint on convolutional layers is tight, multiple recent works try to construct convolution layers with an orthogonal Jacobian matrix (Li et al., 2019b; Trockman & Kolter, 2021; Singla & Feizi, 2021). These approaches avoid the aforementioned issues and allow the training of large, provably 1-Lipschitz CNNs while achieving impressive results.

**Provable defenses against adversarial examples**: A provably robust classifier is one for which we can guarantee that the classifier's prediction remains constant within some region around the input. Most of the existing methods for provable robustness either bound the Lipschitz constant of the neural network or the individual layers (Weng et al., 2018; Zhang et al., 2019; 2018; Wong et al., 2018; Wong & Kolter, 2018; Raghunathan et al., 2018; Croce et al., 2019; Singh et al., 2018; Singla & Feizi, 2020; Zhang et al., 2021; 2022; Wang et al., 2021; Huang et al., 2021). However, these methods do not scale to large and practical networks on the ImageNet dataset (Deng et al., 2009). To scale to such large networks, randomized smoothing (Liu et al., 2018; Cao & Gong, 2017; Lécuyer et al., 2018; Li et al., 2019a; Cohen et al., 2019; Salman et al., 2019; Levine et al., 2019; Kumar et al., 2020a;b) has been proposed as a *probabilistically certified defense*. However, certifying robustness with high probability requires generating a large number of noisy samples leading to high inference-time computation. In contrast, the defense we propose is deterministic and hence not comparable to randomized smoothing. While Levine & Feizi (2021) provide deterministic robustness certificates using randomized smoothing, their certificates are in the $l_1$ norm and not directly applicable for the $l_2$ threat model studied in this work. We discuss the differences between $l_1$ and $l_2$ certificates in Appendix Section C.

## 3 PROBLEM SETUP AND NOTATION

For a vector $\mathbf{v}$, $\mathbf{v}_j$ denotes its $j^{th}$ element. For a matrix $\mathbf{A}$, $\mathbf{A}_{j,:}$ and $\mathbf{A}_{:,k}$ denote the $j^{th}$ row and $k^{th}$ column respectively. Both $\mathbf{A}_{j,:}$ and $\mathbf{A}_{:,k}$ are assumed to be column vectors (thus $\mathbf{A}_{j,:}$ is the transpose of $j^{th}$ row of $\mathbf{A}$). $\mathbf{A}_{j,k}$ denotes the element in $j^{th}$ row and $k^{th}$ column of $\mathbf{A}$. $\mathbf{A}_{:j,:k}$ denotes the matrix containing the first $j$ rows and $k$ columns of $\mathbf{A}$. The same rules are directly extended to higher order tensors. $\mathbf{I}$ denotes the identity matrix, $\mathbb{R}$ to denote the field of real numbers. For $\theta \in \mathbb{R}$, $\mathbf{J}^+(\theta)$ and $\mathbf{J}^-(\theta)$ denote the orthogonal matrices with determinants $+1$ and $-1$ defined as follows:

$$\mathbf{J}^+(\theta) = \begin{bmatrix} \cos\theta & \sin\theta \\ -\sin\theta & \cos\theta \end{bmatrix} \qquad \mathbf{J}^-(\theta) = \begin{bmatrix} \cos\theta & \sin\theta \\ \sin\theta & -\cos\theta \end{bmatrix} \qquad (2)$$

We construct a 1-Lipschitz neural network, $f : \mathbb{R}^d \rightarrow \mathbb{R}^k$ ($d$ is the input dimension, $k$ is the number of classes) by composing 1-Lipschitz convolution layers and GNP activation functions.

To certify robustness for some input $\mathbf{x}$ with prediction $l$, we first define the margin of prediction: $\mathcal{M}_f(\mathbf{x}) = \max\left(0, f_l(\mathbf{x}) - \max_{i \neq l} f_i(\mathbf{x})\right)$ where $f_i(\mathbf{x})$ is the logit for class $i$ and $l$ is the correct label. Using Theorem 7 in Li et al. (2019b), we can derive the robustness certificate (in the $l_2$ norm) as $\mathcal{M}_f(\mathbf{x})/\sqrt{2}$. Thus, the $l_2$ distance of $\mathbf{x}$ to the decision boundary is lower bounded by $\mathcal{M}_f(\mathbf{x})/\sqrt{2}$:

$$\min_{i \neq l} \min_{f_i(\mathbf{x}^*)=f_l(\mathbf{x}^*)} \|\mathbf{x}^* - \mathbf{x}\|_2 \geq \frac{\mathcal{M}_f(\mathbf{x})}{\sqrt{2}} \tag{3}$$

We often use the abbreviation $f_i - f_j : \mathbb{R}^D \to \mathbb{R}$ to denote the function so that:

$$(f_i - f_j)(\mathbf{x}) = f_i(\mathbf{x}) - f_j(\mathbf{x}), \qquad \forall\, \mathbf{x} \in \mathbb{R}^D$$

Our goal is to train the neural network $f$ to achieve the maximum possible provably robust accuracy while also simultaneously improving (or maintaining) standard accuracy.

## 4  LAST LAYER NORMALIZATION

To ensure that the network is 1-Lipschitz so that the certificate in equation (3) is valid, existing 1-Lipschitz neural networks require the weight matrices of all the linear layers of the network to be orthogonal. For the weight matrix in the last layer of the network (that maps the penultimate layer neurons to the logits), $\mathbf{W} \in \mathbb{R}^{k \times m}$ ($k$ is the number of classes, $m$ is the dimension of the penultimate layer, $m > k$), this enforces the following constraints on each row $\mathbf{W}_{i,:} \in \mathbb{R}^m$:

$$\forall j, \ \|\mathbf{W}_{j,:}\|_2 = 1, \qquad \forall i \neq j, \ \mathbf{W}_{j,:} \perp \mathbf{W}_{i,:} \tag{4}$$

Now suppose that for some input $\mathbf{x}$ with label $l$, we want to update $\mathbf{W}$ to increase the logit for the $l^{th}$ class. Since $\|\mathbf{W}_{l,:}\|_2$ is constrained to be 1, the gradient update can only change the direction of the vector $\mathbf{W}_{l,:}$. But now, because the other rows $\{\mathbf{W}_{i,:}, \ i \neq l\}$ are constrained to be orthogonal to $\mathbf{W}_{l,:}$, this further requires an update for all the rows of $\mathbf{W}$. This has the negative effect that during training, any update made to learn some class must necessitate the forgetting of information relevant for the other classes. This can be particularly problematic when the number of classes $k$ is large (such as in CIFAR-100) and thus the number of orthogonality constraints per row i.e. $k-1$ is large.

To address this limitation, first observe that the neural network from the input layer to the penultimate layer (i.e excluding the last linear layer) is 1-Lipschitz. Let $g : \mathbb{R}^d \to \mathbb{R}^m$ be this function so that $f(\mathbf{x}) = \mathbf{W}g(\mathbf{x}) + \mathbf{b}$. This equation suggests that even if $\mathbf{W}$ is not orthogonal, the Lipschitz constant of the function $f_l - f_i$, can be computed by multiplying the Lipschitz constant of $g$ (which is 1) and that of $(\mathbf{W}_{l,:} - \mathbf{W}_{i,:})$ (which is $\|\mathbf{W}_{l,:} - \mathbf{W}_{i,:}\|_2$). The robustness certificate can then be computed as $\mathcal{M}_f(\mathbf{x})/\|\mathbf{W}_{l,:} - \mathbf{W}_{i,:}\|_2$. This procedure leads to the following proposition:

**Proposition 1.** *Given 1-Lipschitz continuous function $g : \mathbb{R}^d \to \mathbb{R}^m$ and $\mathbf{W} \in \mathbb{R}^{k \times m}$, $\mathbf{b} \in \mathbb{R}^k$ ($k$ is the number of classes), construct a new function $f : \mathbb{R}^d \to \mathbb{R}^k$ defined as: $f(\mathbf{x}) = \mathbf{W}g(\mathbf{x}) + \mathbf{b}$. Let $f_l(\mathbf{x}) > \max_{i \neq l} f_i(\mathbf{x})$. The robustness certificate (under the $l_2$ norm) is given by:*

$$\min_{i \neq l} \min_{f_l(\mathbf{x}^*)=f_i(\mathbf{x}^*)} \|\mathbf{x}^* - \mathbf{x}\|_2 \geq \min_{i \neq l} \frac{f_l(\mathbf{x}) - f_i(\mathbf{x})}{\|\mathbf{W}_{l,:} - \mathbf{W}_{i,:}\|_2}$$

Proof is in Appendix Section A.1.

However, in our experiments, we found that using this procedure directly i.e without any constraint on the weight matrix $\mathbf{W}$ often results in large norms of row vectors $\|\mathbf{W}_{i,:}\|_2$, and thus large values of $\|\mathbf{W}_{l,:} - \mathbf{W}_{i,:}\|_2$ and smaller certificates (Theorem 1). To address this problem, we normalize all rows of the matrix to be of unit norm before computing the logits so that for the input $\mathbf{x}$, the logit $g_i(\mathbf{x})$ can be computed as follows:

$$g_i(\mathbf{x}) = \frac{(\mathbf{W}_{i,:})^T f(\mathbf{x})}{\|\mathbf{W}_{i,:}\|_2} + \mathbf{b}_i$$

The robustness certificate can then be computed as follows:

$$\min_{i \neq l} \frac{g_l(\mathbf{x}) - g_i(\mathbf{x})}{\|\mathbf{W}_{l,:}^{(n)} - \mathbf{W}_{i,:}^{(n)}\|_2}, \qquad \text{where } \forall j, \ \mathbf{W}_{j,:}^{(n)} = \frac{\mathbf{W}_{j,:}}{\|\mathbf{W}_{j,:}\|_2}, \ g_j(\mathbf{x}) = \left(\mathbf{W}_{j,:}^{(n)}\right)^T f(\mathbf{x}) + \mathbf{b}_j$$

While each row $\mathbf{W}_{i,:}$ is still constrained to be of unit norm, unlike with orthogonality constraints (equation (4)), their directions are allowed to change freely. This provides significant improvements when the number of classes is large. We call this procedure *Last Layer Normalization* (*LLN*).

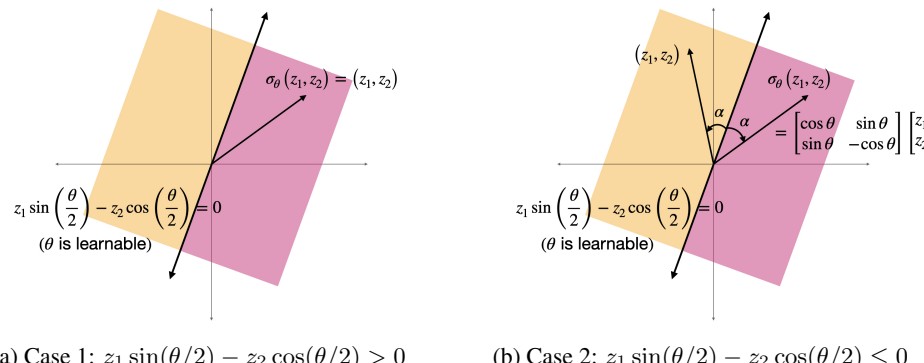

(a) Case 1: $z_1 \sin(\theta/2) - z_2 \cos(\theta/2) > 0$   (b) Case 2: $z_1 \sin(\theta/2) - z_2 \cos(\theta/2) \leq 0$

Figure 1: Illustration of the Householder activation, $\sigma_\theta$. In each colored region, $\sigma_\theta$ is linear. The Jacobian is $\mathbf{I}$ when $(z_1, z_2)$ lies in the pink region (Case 1) and $\mathbf{I} - 2\mathbf{v}\mathbf{v}^T$ in the other region (Case 2) where $\mathbf{v} = [\sin(\theta/2) \quad -\cos(\theta/2)]^T$. Both of these matrices are orthogonal implying $\sigma_\theta$ is GNP.

## 5   CERTIFICATE REGULARIZATION

A limitation of using cross entropy loss for training 1-Lipschitz CNNs is that it is not explicitly designed to maximize the margin $\mathcal{M}_f(\mathbf{x})$ and thus, the robustness certificate. That is, once the cross entropy loss becomes small, the gradients will no longer try to further increase the margin $\mathcal{M}_f(\mathbf{x})$ even though the network may have the capacity to learn bigger margins.

To address this limitation, we can simply *subtract* the certificate i.e $-\mathcal{M}_f(\mathbf{x})/\sqrt{2}$ from the usual cross entropy loss function during training. Observe that we subtract the certificate because we want to *maximize* the certificate values while *minimizing* the cross entropy loss. However, in our experiments we found that this regularization term excessively penalizes the network for the misclassified examples and as a result, the certificate values for the correctly classified inputs are not large. Thus, we propose to use the following regularized loss function during training:

$$\min_{\Omega} \quad \mathbb{E}_{(\mathbf{x},l)\sim D}\left[\ell\left(f_\Omega(\mathbf{x}), l\right) - \gamma \, \mathrm{relu}\left(\frac{\mathcal{M}_f(\mathbf{x})}{\sqrt{2}}\right)\right] \tag{5}$$

In the above equation, $f_\Omega$ denotes the 1-Lipschitz neural network parametrized by $\Omega$, $f_\Omega(\mathbf{x})$ denotes the logits for the input $\mathbf{x}$, $\ell\left(f_\Omega(\mathbf{x}), l\right)$ is the cross entropy loss for input $\mathbf{x}$ with label $l$ and $\gamma > 0$ is the regularization coefficient for maximizing the certificate. We have the minus sign in front of the regularization term $\gamma \, \mathrm{relu}(\mathcal{M}_f(\mathbf{x})/\sqrt{2})$ because we want to *maximize* the certificate while *minimizing* the cross entropy loss. For wrongly classified inputs, $\mathcal{M}_f(\mathbf{x})/\sqrt{2} < 0 \implies \mathrm{relu}(\mathcal{M}_f(\mathbf{x})/\sqrt{2}) = 0$. This ensures that the optimization tries to increase the certificates *only* for the correctly classified inputs. We call the above mentioned procedure *Certificate Regularization* (abbreviated as *CR*).

## 6   HOUSEHOLDER ACTIVATION FUNCTIONS

Recall that given $\mathbf{z} \in \mathbb{R}^m$, the Householder (HH) transformation reflects $\mathbf{z}$ about the hyperplane $\mathbf{v}^T\mathbf{x} = 0$ where $\|\mathbf{v}\|_2 = 1$. The linear transformation is given by the equation $(\mathbf{I} - 2\mathbf{v}\mathbf{v}^T)\mathbf{z}$ where $\mathbf{I} - 2\mathbf{v}\mathbf{v}^T$ is orthogonal because $\|\mathbf{v}\|_2 = 1$. Now, consider the nonlinear function $\sigma_\mathbf{v}$ defined below:

**Definition 1.** *(**Householder Activation of Order** 1) The activation function $\sigma_\mathbf{v} : \mathbb{R}^m \to \mathbb{R}^m$, applied on $\mathbf{z} \in \mathbb{R}^m$, is called the $m$-dimensional Householder Activation of Order 1:*

$$\sigma_\mathbf{v}(\mathbf{z}) = \begin{cases} \mathbf{z}, & \mathbf{v}^T\mathbf{z} > 0, \\ (\mathbf{I} - 2\mathbf{v}\mathbf{v}^T)\mathbf{z}, & \mathbf{v}^T\mathbf{z} \leq 0. \end{cases} \tag{6}$$

Since $\sigma_\mathbf{v}$ is linear when $\mathbf{v}^T\mathbf{z} > 0$ or $\mathbf{v}^T\mathbf{z} < 0$, it is also continuous in both cases. At the hyperplane separating the two cases (i.e., $\mathbf{v}^T\mathbf{z} = 0$) we have: $(\mathbf{I} - 2\mathbf{v}\mathbf{v}^T)\mathbf{z} = \mathbf{z} - 2(\mathbf{v}^T\mathbf{z})\mathbf{v} = \mathbf{z}$ (both linear functions are equal). Thus, $\sigma_\mathbf{v}$ is continuous $\forall \, \mathbf{z} \in \mathbb{R}^m$. Moreover, the Jacobian is either $\mathbf{I}$ or

$\mathbf{I} - 2\mathbf{v}\mathbf{v}^T$ which are both square orthogonal matrices. Thus, $\sigma_{\mathbf{v}}$ is also GNP and 1-Lipschitz. Since these properties hold for all $\mathbf{v}$ satisfying $\|\mathbf{v}\|_2 = 1$, $\mathbf{v}$ can be made a learnable parameter.

While the above arguments suggest that HH transformations are *sufficient* to ensure such functions are continuous, we also prove that they are *necessary*. That is, we prove that if a GNP piecewise linear function $g : \mathbb{R}^m \to \mathbb{R}^m$ transitions between different linear functions $\mathbf{Q}_1\mathbf{z}$ and $\mathbf{Q}_2\mathbf{z}$ (in an open set $S \subset \mathbb{R}^m$) along a hyperplane $\mathbf{v}^T\mathbf{z} = 0$ (where $\|\mathbf{v}\|_2 = 1$), then $g$ is continuous in $S$ *if and only if* $\mathbf{Q}_2 = \mathbf{Q}_1(\mathbf{I} - 2\mathbf{v}\mathbf{v}^T)$. This theoretical result provides a general principle for designing piecewise linear GNP activation functions. The formal result is stated in the following Theorem:

**Theorem 1.** *Given an open set $S \subset \mathbb{R}^m$, orthogonal square matrices $\mathbf{Q}_1 \neq \mathbf{Q}_2$, and vector $\mathbf{v} \in \mathbb{R}^m$ ($\|\mathbf{v}\|_2 = 1$) such that $S \cap \{\mathbf{z} : \mathbf{v}^T\mathbf{z} = 0\} \neq \emptyset$, the function $g$ defined as follows:*

$$g(\mathbf{z}) = \begin{cases} \mathbf{Q}_1\mathbf{z}, & \mathbf{z} \in S, \mathbf{v}^T\mathbf{z} > 0, \\ \mathbf{Q}_2\mathbf{z}, & \mathbf{z} \in S, \mathbf{v}^T\mathbf{z} \leq 0 \end{cases} \tag{7}$$

*is continuous in $S$ if and only if $\mathbf{Q}_2 = \mathbf{Q}_1(\mathbf{I} - 2\mathbf{v}\mathbf{v}^T)$.*

Proof of Theorem 1 is in Appendix A.2. Note that since the matrix $\mathbf{I} - 2\mathbf{v}\mathbf{v}^T$ has determinant $-1$, the above theorem necessitates that $\det(\mathbf{Q}_1) = -\det(\mathbf{Q}_2)$ i.e the determinant of the Jacobian must change sign whenever the Jacobian of a piecewise linear GNP activation function changes.

Recall that for the $\mathrm{MaxMin}$ activation function, $\mathrm{MaxMin}(z_1, z_2) = (z_1, z_2)$ if $z_1 > z_2$ and $(z_2, z_1)$ otherwise. Thus, the Jacobian of $\mathrm{MaxMin}$ for $z_1 > z_2$ case is $\mathbf{I} = \mathbf{J}^+(0)$ while for $z_1 \leq z_2$ is $\mathbf{J}^-(\pi/2)$. Using Theorem 1, we can easily prove that $\mathrm{MaxMin}$ is a special case of the more general Householder activation functions where the Jacobian $\mathbf{J}^-(\pi/2)$ is replaced with $\mathbf{J}^-(\theta)$ and the conditions $z_1 > z_2$ are replaced with $z_1 \sin(\theta/2) > z_2 \cos(\theta/2)$ (similarly for $\leq$). The construction of Householder activations in 2 dimensions, denoted by $\sigma_\theta$, is given in the following corollary:

**Corollary 1.** *The function $\sigma_\theta : \mathbb{R}^2 \to \mathbb{R}^2$ defined as*

$$\sigma_\theta(z_1, z_2) = \begin{cases} \begin{bmatrix} 1 & 0 \\ 0 & 1 \end{bmatrix} \begin{bmatrix} z_1 \\ z_2 \end{bmatrix} & \text{if } z_1 \sin(\theta/2) - z_2 \cos(\theta/2) > 0 \\ \begin{bmatrix} \cos\theta & \sin\theta \\ \sin\theta & -\cos\theta \end{bmatrix} \begin{bmatrix} z_1 \\ z_2 \end{bmatrix} & \text{if } z_1 \sin(\theta/2) - z_2 \cos(\theta/2) \leq 0 \end{cases} \tag{8}$$

*is continuous and is called* 2D *Householder Activation of Order* 1.

The two cases are demonstrated in Figure 1a and Figure 1b, respectively. Since $\sigma_\theta$ is continuous, GNP and 1-Lipschitz $\forall\, \theta \in \mathbb{R}$, $\theta$ is a learnable parameter. For $\theta = \pi/2$ in equation (8), $\sigma_\theta$ is equivalent to $\mathrm{MaxMin}$. Thus, $\sigma_\theta$ is at least as expressive as $\mathrm{MaxMin}$.

A major limitation of using $\sigma_{\mathbf{v}}$ (equation (6)) directly is that it has only 2 linear regions and is thus limited in its expressive power. In contrast, $\mathrm{MaxMin}$ first divides the preactivation $\mathbf{z} \in \mathbb{R}^m$ (assuming $m$ is divisible by 2) into $m/2$ groups of size 2 each. Since each group has 2 linear regions, we get $2^{m/2}$ linear regions from the $m/2$ groups. Thus, to increase the expressive power, we similarly divide $\mathbf{z}$ into $m/2$ groups of size 2 each and apply the 2-dimensional Householder activation function of Order 1 ($\sigma_\theta$) to each group resulting in $2^{m/2}$ linear regions (same as $\mathrm{MaxMin}$).

To apply $\sigma_\theta$ to the output of a convolution layer $\mathbf{z} \in \mathbb{R}^{m \times n \times n}$ ($m$ is the number of channels and $n \times n$ is the spatial size), we first split $\mathbf{z}$ into 2 tensors along the channel dimension giving the tensors: $\mathbf{z}_{:m/2,:,:}$ and $\mathbf{z}_{m/2:,:,:}$. Each of these tensors is of size $m/2 \times n \times n$ giving $n^2 m/2$ groups. We use the same $\theta$ for each pair of channels (irrespective of spatial location) resulting in $m/2$ learnable parameters. We initialize each $\theta = \pi/2$ so that $\sigma_\theta$ is equivalent to $\mathrm{MaxMin}$ at initialization.

## 7 EXPERIMENTS

Our goal is to evaluate the effectiveness of the three changes proposed in this work: (a) Last Layer Normalization, (b) Certificate regularization and (c) Householder activation functions. We perform experiments under the setting of provably robust image classification on CIFAR-10 and CIFAR-100 datasets using the same 1-Lipschitz CNN architectures used by Singla & Feizi (2021) (LipConvnet-5, 10, 15, ..., 40) due to their superior performance over the prior works. We compare with the three

orthogonal convolution layers in the literature: SOC (Singla & Feizi, 2021), BCOP (Li et al., 2019b) and Cayley (Trockman & Kolter, 2021) using $\mathrm{MaxMin}$ as the activation function.

We use SOC with $\mathrm{MaxMin}$ as the primary baseline for comparison in the maintext due to their superior performance over the prior works (BCOP, Cayley). Results using BCOP and Cayley convolutions are given in Appendix Sections H and I for completeness. We use the same implementations for these convolution layers as given in their respective github repositories. We compare the provable robust accuracy using 3 different values of the $l_2$ perturbation radius $\rho = 36/255,\ 72/255,\ 108/255$. In both Tables 1 and 2, for all networks, we use SOC as the convolution layer. Using SOC, we achieve the same bound on the approximation error of an orthogonal Jacobian as achieved in Singla & Feizi (2021) i.e. $2.42 \times 10^{-6}$ across all convolution layers in all networks. Thus, even for the network with 40 layers, this results in maximum Lipschitz constant of $(1 + 2.42 \times 10^{-6})^{40} = 1.000097 \leq 1.0001$. Thus, the Lipschitz constant across all our networks is bounded by 1.0001. The symbol HH (in Tables 1, 2) is for the 2D Householder Activation of order 1 or $\sigma_\theta$ (defined in equation (8)).

All experiments were performed using 1 NVIDIA GeForce RTX 2080 Ti GPU. All networks were trained for 200 epochs with initial learning rate of 0.1, dropped by a factor of 0.1 after 100 and 150 epochs. For Certificate Regularization (or CR), we set the parameter $\gamma = 0.5$.

## 7.1 RESULTS ON CIFAR-100

In Table 1, for each architecture, the row "SOC + $\mathrm{MaxMin}$" uses the $\mathrm{MaxMin}$ activation, "+ LLN" adds Last Layer Normalization (uses $\mathrm{MaxMin}$), "+ HH" replaces $\mathrm{MaxMin}$ with $\sigma_\theta$ (also uses LLN) , "+ CR" also adds Certificate Regularization with $\gamma = 0.1$ (uses both $\sigma_\theta$ and LLN). The column, "Increase (Standard)" denotes the increase in standard accuracy relative to "SOC + $\mathrm{MaxMin}$".

By adding LLN (the row "+ LLN"), we observe gains in standard (min gain of $1.10\%$) and provable robust accuracy (min gain of $1.71\%$ at $\rho = 36/255$) across all the LipConvnet architectures (gains relative to "SOC + $\mathrm{MaxMin}$"). These gains are smallest for the LipConvnet-40 network with the maximum depth. However, replacing $\mathrm{MaxMin}$ with the $\sigma_\theta$ activation further improves the standard (min gain of $3.65\%$) and provable robust accuracy (min gain of $4.46\%$ at $\rho = 36/255$) across all networks (again relative to "SOC + $\mathrm{MaxMin}$"). We observe that replacing $\mathrm{MaxMin}$ with $\sigma_\theta$ significantly improves the performance of the deeper LipConvnet-35, 40 networks.

Adding CR further improves the provable robust accuracy while only slightly reducing the standard accuracy. Because LLN significantly improves the standard accuracy, we compare the standard accuracy numbers between rows "+ CR" and "+ LLN" to evaluate the drop due to CR. We observe a small drop in standard accuracy ($-0.04\%, -0.11\%$) only for LipConvnet-5 and LipConvnet-15 networks. For the other networks, the standard accuracy actually *increases*.

## 7.2 RESULTS ON CIFAR-10

In Table 2, for each architecture, the row "SOC + $\mathrm{MaxMin}$" uses the $\mathrm{MaxMin}$ activation, the row "+ HH" uses $\sigma_\theta$ activation (replacing $\mathrm{MaxMin}$) and the row "+ CR" also adds Certificate Regularization with $\gamma = 0.1$ (again using $\sigma_\theta$ as the activation). Due to the small number of classes in CIFAR-10, we do not observe significant gains using Last Layer Normalization or LLN (Appendix Table 10). Thus, we do not include LLN for any of the results in Table 2. The column, "Increase $(108/255)$" denotes the increase in provable robust accuracy with $\rho = 108/255$ relative to "SOC + $\mathrm{MaxMin}$".

For LipConvnet-25, 30, 35, 40 architectures, we observe gains in both the standard and provable robust accuracy by replacing $\mathrm{MaxMin}$ with the HH activation (i.e $\sigma_\theta$). Similar to what we observe for CIFAR-100 in Table 1, the gains in provable robust accuracy ($\rho = 108/255$) are significantly higher for deeper networks: LipConvnet-35 ($3.65\%$) and LipConvnet-40 ($4.35\%$) with decent gains in standard accuracy ($1.71$ and $1.61\%$ respectively).

Again similar to CIFAR-100, adding CR further boosts the provable robust accuracy while slightly reducing the standard accuracy. Comparing "+ CR" with "SOC + $\mathrm{MaxMin}$", we observe small drops in standard accuracy for LipConvnet-5, 10, ..., 30 networks (max. drop of $-0.56\%$), and gains for LipConvnet-35 ($+0.52\%$) and LipConvnet-40 ($+0.96\%$). For provable robust accuracy ($\rho = 108/255$), we observe very significant gains of $> 4.96\%$ for all networks and $> 8\%$ for the deeper LipConvnet-35, 40 networks.

| Architecture | Methods | Standard Accuracy | Provable Robust Acc. ($\rho =$) | | | Increase |
| | | | 36/255 | 72/255 | 108/255 | (Standard) |
|---|---|---|---|---|---|---|
| LipConvnet-5 | SOC + MaxMin | 42.71% | 27.86% | 17.45% | 9.99% | _ |
| | + LLN | 45.86% | 31.93% | 21.17% | 13.21% | +3.15% |
| | + HH | **46.36%** | 32.64% | 21.19% | 13.12% | **+3.65%** |
| | + CR | 45.82% | **32.99%** | **22.48%** | **14.79%** | +3.11% |
| LipConvnet-10 | SOC + MaxMin | 43.72% | 29.39% | 18.56% | 11.16% | _ |
| | + LLN | 46.88% | 33.32% | 22.08% | 13.87% | +3.16% |
| | + HH | **47.96%** | 34.30% | 22.35% | 14.48% | **+4.24%** |
| | + CR | 47.07% | **34.53%** | **23.50%** | **15.66%** | +3.35% |
| LipConvnet-15 | SOC + MaxMin | 42.92% | 28.81% | 17.93% | 10.73% | _ |
| | + LLN | 47.72% | 33.52% | 21.89% | 13.76% | +4.80% |
| | + HH | **47.72%** | 33.97% | 22.45% | 13.81% | **+4.80%** |
| | + CR | 47.61% | **34.54%** | **23.16%** | **15.09%** | +4.69% |
| LipConvnet-20 | SOC + MaxMin | 43.06% | 29.34% | 18.66% | 11.20% | _ |
| | + LLN | 46.86% | 33.48% | 22.14% | 14.10% | +3.80% |
| | + HH | 47.71% | 34.22% | 22.93% | 14.57% | +4.65% |
| | + CR | **47.84%** | **34.77%** | **23.70%** | **15.84%** | **+4.78%** |
| LipConvnet-25 | SOC + MaxMin | 43.37% | 28.59% | 18.18% | 10.85% | _ |
| | + LLN | 46.32% | 32.87% | 21.53% | 13.86% | +2.95% |
| | + HH | **47.70%** | 34.00% | 22.67% | 14.57% | **+4.33%** |
| | + CR | 46.87% | **34.09%** | **23.41%** | **15.61%** | +3.50% |
| LipConvnet-30 | SOC + MaxMin | 42.87% | 28.74% | 18.47% | 11.21% | _ |
| | + LLN | 46.18% | 32.82% | 21.52% | 13.52% | +3.31% |
| | + HH | 46.80% | 33.72% | 22.70% | 14.31% | +3.93% |
| | + CR | **46.92%** | **34.17%** | **23.21%** | **15.84%** | **+4.05%** |
| LipConvnet-35 | SOC + MaxMin | 42.42% | 28.34% | 18.10% | 10.96% | _ |
| | + LLN | 45.22% | 32.10% | 21.28% | 13.25% | +2.80% |
| | + HH | 46.21% | 32.80% | 21.55% | 14.13% | +3.79% |
| | + CR | **46.88%** | **33.64%** | **23.34%** | **15.73%** | **+4.46%** |
| LipConvnet-40 | SOC + MaxMin | 41.84% | 28.00% | 17.40% | 10.28% | _ |
| | + LLN | 42.94% | 29.71% | 19.30% | 11.99% | +1.10% |
| | + HH | **45.84%** | **32.79%** | 21.98% | 14.07% | **+4.00%** |
| | + CR | 45.03% | 32.57% | **22.37%** | **14.76%** | +3.19% |

Table 1: Results for provable robustness against adversarial examples on the CIFAR-100 dataset. Results with BCOP and Cayley convolutions are in Appendix Tables 13 and 14.

## 8 CONCLUSION

In this work, we introduce a procedure to certify robustness of 1-Lipschitz convolutional neural networks without orthogonalizing of the last linear layer of the network. Additionally, we introduce a certificate regularization that significantly improves the provable robust accuracy for these models at higher $l_2$ radii. Finally, we introduce a class of GNP activation functions called Householder (or HH) activations and prove that the Jacobian of any Gradient Norm Preserving (GNP) piecewise linear function is only allowed to change via Householder transformations for the function to be continuous

| Architecture | Methods | Standard Accuracy | Provable Robust Acc. ($\rho =$) | | | Increase |
| | | | 36/255 | 72/255 | 108/255 | (108/255) |
|---|---|---|---|---|---|---|
| LipConvnet-5 | SOC + MaxMin | 75.78% | 59.18% | 42.01% | 27.09% | _ |
| | **+ HH** | **76.30%** | 60.12% | 43.20% | 27.39% | +0.30% |
| | **+ CR** | 75.31% | **60.37%** | **45.62%** | **32.38%** | **+5.29%** |
| LipConvnet-10 | SOC + MaxMin | 76.45% | 60.86% | 44.15% | 29.15% | _ |
| | **+ HH** | **76.86%** | 61.52% | 44.91% | 29.90% | +0.75% |
| | **+ CR** | 76.23% | **62.57%** | **47.70%** | **34.15%** | **+5.00%** |
| LipConvnet-15 | SOC + MaxMin | 76.68% | 61.36% | 44.28% | 29.66% | _ |
| | **+ HH** | **77.41%** | 61.92% | 45.60% | 31.10% | +1.44% |
| | **+ CR** | 76.39% | **62.96%** | **48.47%** | **35.47%** | **+5.81%** |
| LipConvnet-20 | SOC + MaxMin | 76.90% | 61.87% | 45.79% | 31.08% | _ |
| | **+ HH** | **76.99%** | 61.76% | 45.59% | 30.99% | -0.09% |
| | **+ CR** | 76.34% | **62.63%** | **48.69%** | **36.04%** | **+4.96%** |
| LipConvnet-25 | SOC + MaxMin | 75.24% | 60.17% | 43.55% | 28.60% | _ |
| | **+ HH** | **76.37%** | 61.50% | 44.72% | 29.83% | +1.23% |
| | **+ CR** | 75.21% | **61.98%** | **47.93%** | **34.92%** | **+6.32%** |
| LipConvnet-30 | SOC + MaxMin | 74.51% | 59.06% | 42.46% | 28.05% | _ |
| | **+ HH** | **75.25%** | 59.90% | 43.85% | 29.35% | +1.30% |
| | **+ CR** | 74.23% | **60.64%** | **46.51%** | **34.08%** | **+6.03%** |
| LipConvnet-35 | SOC + MaxMin | 73.73% | 58.50% | 41.75% | 27.20% | _ |
| | **+ HH** | **75.44%** | 61.05% | 45.38% | 30.85% | +3.65% |
| | **+ CR** | 74.25% | **61.30%** | **47.60%** | **35.21%** | **+8.01%** |
| LipConvnet-40 | SOC + MaxMin | 71.63% | 54.39% | 37.92% | 24.13% | _ |
| | **+ HH** | **73.24%** | 58.12% | 42.24% | 28.48% | +4.35% |
| | **+ CR** | 72.59% | **59.04%** | **44.92%** | **32.87%** | **+8.74%** |

Table 2: Results for provable robustness against adversarial examples on the CIFAR-10 dataset. Results with BCOP and Cayley convolutions are in Appendix Tables 7 and 8.

which provides a general principle for designing piecewise linear GNP functions. These ideas lead to improved deterministic $\ell_2$ robustness certificates on CIFAR-10 and CIFAR-100.

## ACKNOWLEDGEMENTS

This project was supported in part by NSF CAREER AWARD 1942230, a grant from NIST 60NANB20D134, HR001119S0026-GARD-FP-052, HR00112090132, ONR YIP award N00014-22-1-2271, Army Grant W911NF2120076.

## REPRODUCIBILITY

The code for reproducing all experiments in the paper is available at `https://github.com/singlasahil14/SOC`.

## ETHICS STATEMENT

This paper introduces novel set of techniques for improving the provable robustness of 1-Lipschitz CNNs. We do not see any foreseeable negative consequences associated with our work.

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

# A  PROOFS

## A.1  PROOF OF PROPOSITION 1

*Proof.* We proceed by computing the Lipschitz constant of the function $f_l - f_i$.
The gradient of the function: $f_l - f_i$ at $\mathbf{x}$ can be computed using the chain rule:

$$\nabla_{\mathbf{x}} (f_l - f_i) = (\mathbf{W}_{l,:} - \mathbf{W}_{i,:})^T \nabla_{\mathbf{x}} g$$

Since $g$ is given to be 1-Lipschitz, the Lipschitz constant of $f_l - f_i$ can be computed using the above equation as follows:

$$\|\nabla_{\mathbf{x}} (f_l - f_i)\|_2 \leq \| (\mathbf{W}_{l,:} - \mathbf{W}_{i,:})^T (\nabla_{\mathbf{x}} g)\|_2$$
$$\|\nabla_{\mathbf{x}} (f_l - f_i)\|_2 \leq \|\mathbf{W}_{l,:} - \mathbf{W}_{i,:}\|_2 \|\nabla_{\mathbf{x}} g\|_2 \leq \|\mathbf{W}_{l,:} - \mathbf{W}_{i,:}\|_2$$

Thus, the distance of $\mathbf{x}$ to the decision boundary $f_l - f_i = 0$, is lower bounded by:

$$\min_{f_l(\mathbf{x}^*)=f_i(\mathbf{x}^*)} \|\mathbf{x}^* - \mathbf{x}\|_2 \geq \frac{f_l(\mathbf{x}) - f_i(\mathbf{x})}{\|\mathbf{W}_{l,:} - \mathbf{W}_{i,:}\|_2}$$

Thus, the distance to decision boundary across all classes $i \neq l$ is lower bounded by:

$$\min_{i \neq l} \min_{f_l(\mathbf{x}^*)=f_i(\mathbf{x}^*)} \|\mathbf{x}^* - \mathbf{x}\|_2 \geq \min_{i \neq l} \frac{f_l(\mathbf{x}) - f_i(\mathbf{x})}{\|\mathbf{W}_{l,:} - \mathbf{W}_{i,:}\|_2}$$

$\square$

## A.2  PROOF OF THEOREM 1

*Proof.* We first prove that if $\mathbf{Q}_2 = (\mathbf{I} - 2\mathbf{v}\mathbf{v}^T)\mathbf{Q}_1$, then the function $g$ is continuous.
First, observe that for $\mathbf{v}^T\mathbf{z} > 0$, $g(\mathbf{z}) = \mathbf{Q}_1\mathbf{z}$ which is continuous.
Similarly, for $\mathbf{v}^T\mathbf{z} < 0$, $g(\mathbf{z}) = \mathbf{Q}_2\mathbf{z}$ which is again continuous.
This proves that the function $g$ is continuous when $\mathbf{v}^T\mathbf{z} > 0$ or $\mathbf{v}^T\mathbf{z} < 0$.
Thus, to prove continuity $\forall \mathbf{z} \in S$, we must prove that:

$$\mathbf{Q}_1\mathbf{z} = \mathbf{Q}_2\mathbf{z} \qquad \forall \mathbf{z} : \mathbf{v}^T\mathbf{z} = 0 \tag{9}$$

Since $\mathbf{Q}_2 = \mathbf{Q}_1(\mathbf{I} - 2\mathbf{v}\mathbf{v}^T)$, we have:

$$\mathbf{Q}_2 - \mathbf{Q}_1 = -2\mathbf{Q}_1\mathbf{v}\mathbf{v}^T$$
$$(\mathbf{Q}_2 - \mathbf{Q}_1)\mathbf{z} = -2(\mathbf{Q}_1\mathbf{v}\mathbf{v}^T)\mathbf{z} = -2\mathbf{Q}_1\mathbf{v}(\mathbf{v}^T\mathbf{z})$$

The above equation directly proves (9).

Now, we prove the other direction i.e if $g$ is continuous in $S$ then, $\mathbf{Q}_2 = \mathbf{Q}_1(\mathbf{I} - 2\mathbf{v}\mathbf{v}^T)$.
Since $g$ is continuous for all $\mathbf{z} : \mathbf{v}^T\mathbf{z} = 0$, we have:

$$\mathbf{Q}_2\mathbf{z} = \mathbf{Q}_1\mathbf{z} \qquad \forall \mathbf{z} : \mathbf{v}^T\mathbf{z} = 0$$
$$(\mathbf{Q}_2 - \mathbf{Q}_1)\mathbf{z} = \mathbf{0} \qquad \forall \mathbf{z} : \mathbf{v}^T\mathbf{z} = 0$$

Since $\mathbf{z} \in \mathbb{R}^m$, we know that the set of vectors: $\{\mathbf{z} : \mathbf{v}^T\mathbf{z} = 0\}$ spans a $m - 1$ dimensional subspace.
Thus, the null space of $\mathbf{Q}_2 - \mathbf{Q}_1$ is of size $m - 1$.
This in turn implies that $\mathbf{Q}_2 - \mathbf{Q}_1$ is a rank one matrix given by the following equation:

$$\mathbf{Q}_2 - \mathbf{Q}_1 = \mathbf{u}\mathbf{v}^T \tag{10}$$

where the vector $\mathbf{u}$ remains to be determined.
Since $\mathbf{Q}_1$ and $\mathbf{Q}_2$ are orthogonal matrices, we have the following set of equations:

$$\mathbf{Q}_2^T\mathbf{Q}_2 = (\mathbf{Q}_1 + \mathbf{u}\mathbf{v}^T)^T (\mathbf{Q}_1 + \mathbf{u}\mathbf{v}^T) \tag{11}$$
$$\mathbf{Q}_2\mathbf{Q}_2^T = (\mathbf{Q}_1 + \mathbf{u}\mathbf{v}^T) (\mathbf{Q}_1 + \mathbf{u}\mathbf{v}^T)^T \tag{12}$$

We first simplify equation (11):

$$\mathbf{Q}_2^T \mathbf{Q}_2 = \left(\mathbf{Q}_1^T + \mathbf{v}\mathbf{u}^T\right)\left(\mathbf{Q}_1 + \mathbf{u}\mathbf{v}^T\right)$$
$$\mathbf{I} = \mathbf{I} + \mathbf{v}\left(\mathbf{Q}_1^T\mathbf{u}\right)^T + \left(\mathbf{Q}_1^T\mathbf{u}\right)\mathbf{v}^T + \left(\mathbf{u}^T\mathbf{u}\right)\mathbf{v}\mathbf{v}^T$$
$$0 = \mathbf{v}\left(\mathbf{Q}_1^T\mathbf{u}\right)^T + \left(\mathbf{Q}_1^T\mathbf{u}\right)\mathbf{v}^T + \left(\mathbf{u}^T\mathbf{u}\right)\mathbf{v}\mathbf{v}^T$$
$$- \left(\mathbf{u}^T\mathbf{u}\right)\mathbf{v}\mathbf{v}^T = \mathbf{v}\left(\mathbf{u}^T\mathbf{Q}_1\right) + \left(\mathbf{Q}_1^T\mathbf{u}\right)\mathbf{v}^T$$

Right multiplying both sides by $\mathbf{v}$ and using $\|\mathbf{v}\|_2 = 1$, we get:

$$- \left(\mathbf{u}^T\mathbf{u}\right)\mathbf{v} = \left(\mathbf{u}^T\mathbf{Q}_1\mathbf{v}\right)\mathbf{v} + \mathbf{Q}_1^T\mathbf{u}$$
$$\mathbf{Q}_1^T\mathbf{u} = -\left(\mathbf{u}^T\mathbf{u} + \mathbf{u}^T\mathbf{Q}_1\mathbf{v}\right)\mathbf{v} = \lambda\mathbf{v}$$
$$\mathbf{u} = \lambda\mathbf{Q}_1\mathbf{v}, \quad \text{where } \lambda = -\left(\mathbf{u}^T\mathbf{u} + \mathbf{u}^T\mathbf{Q}_1\mathbf{v}\right) \tag{13}$$

Substituting $\mathbf{u}$ using equation (13) in equation (10), we get:

$$\mathbf{Q}_2 - \mathbf{Q}_1 = \lambda\mathbf{Q}_1\mathbf{v}\mathbf{v}^T$$
$$\mathbf{Q}_2 = \mathbf{Q}_1\left(\mathbf{I} + \lambda\mathbf{v}\mathbf{v}^T\right)$$

Since $\mathbf{Q}_2^T\mathbf{Q}_2 = \mathbf{I}$, we have:

$$\mathbf{Q}_2^T\mathbf{Q}_2 = \left(\mathbf{Q}_1\left(\mathbf{I} + \lambda\mathbf{v}\mathbf{v}^T\right)\right)^T\mathbf{Q}_1\left(\mathbf{I} + \lambda\mathbf{v}\mathbf{v}^T\right)$$
$$\mathbf{Q}_2^T\mathbf{Q}_2 = \left(\mathbf{I} + \lambda\mathbf{v}\mathbf{v}^T\right)\mathbf{Q}_1^T\mathbf{Q}_1\left(\mathbf{I} + \lambda\mathbf{v}\mathbf{v}^T\right)$$
$$\mathbf{I} = \left(\mathbf{I} + \lambda\mathbf{v}\mathbf{v}^T\right)\left(\mathbf{I} + \lambda\mathbf{v}\mathbf{v}^T\right)$$
$$\mathbf{I} = \mathbf{I} + 2\lambda\mathbf{v}\mathbf{v}^T + \lambda^2\mathbf{v}\mathbf{v}^T$$
$$\implies \lambda = 0 \text{ or } \lambda = -2$$

Since $\lambda = 0$ would imply $\mathbf{Q}_1 = \mathbf{Q}_2$ which is not allowed by the assumption of the Theorem that $\mathbf{Q}_1 \neq \mathbf{Q}_2$.
$\lambda = -2$ is the only possibility allowed.
This proves the other direction i.e:

$$\mathbf{Q}_2 = \mathbf{Q}_1\left(\mathbf{I} - 2\mathbf{v}\mathbf{v}^T\right)$$

$\square$

### A.3 PROOF OF COROLLARY 1

*Proof.* Subsitute $\mathbf{Q}_1, \mathbf{v}$ as follows in Theorem 1.

$$\mathbf{Q}_1 = \begin{bmatrix} 1 & 0 \\ 0 & 1 \end{bmatrix}$$
$$\mathbf{v} = \begin{bmatrix} +\sin(\theta/2) \\ -\cos(\theta/2) \end{bmatrix}$$
$$\mathbf{Q}_2 = \mathbf{I} - 2\mathbf{v}\mathbf{v}^T$$
$$\mathbf{Q}_2 = \begin{bmatrix} 1 & 0 \\ 0 & 1 \end{bmatrix} - 2\begin{bmatrix} \sin(\theta/2) \\ -\cos(\theta/2) \end{bmatrix}\begin{bmatrix} \sin(\theta/2) & -\cos(\theta/2) \end{bmatrix}$$
$$\mathbf{Q}_2 = \begin{bmatrix} 1 & 0 \\ 0 & 1 \end{bmatrix} - 2\begin{bmatrix} \sin^2(\theta/2) & -\sin(\theta/2)\cos(\theta/2) \\ -\sin(\theta/2)\cos(\theta/2) & \cos^2(\theta/2) \end{bmatrix}$$
$$\mathbf{Q}_2 = \begin{bmatrix} 1 - 2\sin^2(\theta/2) & 2\sin(\theta/2)\cos(\theta/2) \\ 2\sin(\theta/2)\cos(\theta/2) & 1 + 2\cos^2(\theta/2) \end{bmatrix}$$
$$\mathbf{Q}_2 = \begin{bmatrix} \cos(\theta) & \sin(\theta) \\ \sin(\theta) & -\cos(\theta) \end{bmatrix}$$

Theorem 1 then directly implies the corollary. $\square$

## A.4 Proof of Theorem 2

**Theorem 2.** *Given:* $0 \le \theta_0 < \theta_1 \cdots < \theta_{2n} = 2\pi + \theta_0$ *such that* $\sum_{i=0}^{n-1}(\theta_{2i+1} - \theta_{2i}) = \pi$ *and* $\alpha_i = 2\sum_{j=0}^{i} \theta_{i-j}(-1)^j$, *The function* $\sigma_\Theta : \mathbb{R}^2 \to \mathbb{R}^2$ *is continuous, GNP and 1-Lipschitz where* $\Theta = [\theta_0, \theta_1, \ldots, \theta_{2n-1}]$ *(also called* 2D *Householder Activation of order* $n$):

$$\sigma_\Theta(z_1, z_2) = \begin{bmatrix} \cos\alpha_i & \sin\alpha_i \\ (-1)^i \sin\alpha_i & (-1)^{i+1}\cos\alpha_i \end{bmatrix} \begin{bmatrix} z_1 \\ z_2 \end{bmatrix} \qquad \theta_i \le \varphi < \theta_{i+1} \qquad (14)$$

*where* $\varphi \in [\theta_0, \ \theta_{2n} = 2\pi + \theta_0)$ *and* $\cos(\varphi) = z_1/\sqrt{z_1^2 + z_2^2}, \ \sin(\varphi) = z_2/\sqrt{z_1^2 + z_2^2}$.

*Proof.* We are given the following:

$$\sum_{i=0}^{n-1}(\theta_{2i+1} - \theta_{2i}) = \pi, \qquad \alpha_i = 2\sum_{j=0}^{i}\theta_{i-j}(-1)^j \qquad (15)$$

Note that by definition (equation (14)), the function is linear for $\theta_i < \varphi < \theta_{i+1}$ and hence continuous.

Furthermore, since $\varphi \in [\theta_0, \ \theta_{2n})$, we proceed to prove continuity for the following two cases:

**Case 1:** $\theta_i - \epsilon < \varphi < \theta_i + \epsilon, \qquad \epsilon > 0, \ i \ge 1$

**Case 2:** $\theta_0 < \varphi < \theta_0 + \epsilon, \qquad \theta_{2n} - \epsilon < \varphi < \theta_{2n}, \quad \epsilon > 0$

**Proof for Case 1:**

Using equation (14), we know that $\sigma_\Theta$ realizes different linear functions for $\theta_i - \epsilon < \varphi < \theta_i$ and $\theta_i < \varphi < \theta_i + \epsilon$.

Thus, for $\sigma_\Theta$ to be continuous, we require that the two linear functions be the same at the boundary i.e $\varphi = \theta_i$.

We first write the input $(z_1, \ z_2)$ in terms of shifted polar coordinates i.e: $(r\cos(\varphi), \ r\sin(\varphi))$ where $r = \sqrt{z_1^2 + z_2^2}$ and $\cos\varphi = z_1/\sqrt{z_1^2 + z_2^2}, \ \sin\varphi = z_2/\sqrt{z_1^2 + z_2^2}, \varphi \in [\theta_0, \theta_0 + 2\pi)$

Thus, the function for $\theta_i - \epsilon < \varphi < \theta_i$ is given by:

$$\sigma_\Theta(r\cos\varphi, r\sin\varphi) = \begin{bmatrix} \cos\alpha_{i-1} & \sin\alpha_{i-1} \\ (-1)^{i-1}\sin\alpha_{i-1} & (-1)^i\cos\alpha_{i-1} \end{bmatrix} \begin{bmatrix} r\cos\varphi \\ r\sin\varphi \end{bmatrix} \qquad (16)$$

Similarly, the function for $\theta_i < \varphi < \theta_i + \epsilon$ is given by:

$$\sigma_\Theta(r\cos\varphi, r\sin\varphi) = \begin{bmatrix} \cos\alpha_i & \sin\alpha_i \\ (-1)^i\sin\alpha_i & (-1)^{i+1}\cos\alpha_i \end{bmatrix} \begin{bmatrix} r\cos\varphi \\ r\sin\varphi \end{bmatrix} \qquad (17)$$

The difference in the function values at the boundary i.e $\varphi = \theta_i$, obtained by subtracting equations (17) and (16) is given as follows:

$$\begin{bmatrix} \cos\alpha_i & \sin\alpha_i \\ (-1)^i\sin\alpha_i & (-1)^{i+1}\cos\alpha_i \end{bmatrix} \begin{bmatrix} r\cos\theta_i \\ r\sin\theta_i \end{bmatrix} - \begin{bmatrix} \cos\alpha_{i-1} & \sin\alpha_{i-1} \\ (-1)^{i-1}\sin\alpha_{i-1} & (-1)^i\cos\alpha_{i-1} \end{bmatrix} \begin{bmatrix} r\cos\theta_i \\ r\sin\theta_i \end{bmatrix}$$

$$= r\begin{bmatrix} \cos\alpha_i - \cos\alpha_{i-1} & \sin\alpha_i - \sin\alpha_{i-1} \\ (-1)^i\sin\alpha_i - (-1)^{i-1}\sin\alpha_{i-1} & (-1)^{i+1}\cos\alpha_i - (-1)^i\cos\alpha_{i-1} \end{bmatrix} \begin{bmatrix} \cos\theta_i \\ \sin\theta_i \end{bmatrix}$$

$$= r\begin{bmatrix} \cos\alpha_i - \cos\alpha_{i-1} & \sin\alpha_i - \sin\alpha_{i-1} \\ (-1)^i(\sin\alpha_i + \sin\alpha_{i-1}) & (-1)^{i+1}(\cos\alpha_i + \cos\alpha_{i-1}) \end{bmatrix} \begin{bmatrix} \cos\theta_i \\ \sin\theta_i \end{bmatrix}$$

Using sum-to-product trigonometric identities, the above equals:

$$= r\begin{bmatrix} 2\sin\left(\frac{\alpha_{i-1}-\alpha_i}{2}\right)\sin\left(\frac{\alpha_{i-1}+\alpha_i}{2}\right) & 2\sin\left(\frac{\alpha_i-\alpha_{i-1}}{2}\right)\cos\left(\frac{\alpha_i+\alpha_{i-1}}{2}\right) \\ 2(-1)^i\sin\left(\frac{\alpha_i+\alpha_{i-1}}{2}\right)\cos\left(\frac{\alpha_i-\alpha_{i-1}}{2}\right) & 2(-1)^{i+1}\cos\left(\frac{\alpha_{i-1}-\alpha_i}{2}\right)\cos\left(\frac{\alpha_{i-1}+\alpha_i}{2}\right) \end{bmatrix} \begin{bmatrix} \cos\theta_i \\ \sin\theta_i \end{bmatrix}$$

$$= 2r\begin{bmatrix} \sin\left(\frac{\alpha_{i-1}-\alpha_i}{2}\right)\sin\left(\frac{\alpha_{i-1}+\alpha_i}{2}\right) & -\sin\left(\frac{\alpha_{i-1}-\alpha_i}{2}\right)\cos\left(\frac{\alpha_i+\alpha_{i-1}}{2}\right) \\ (-1)^i\sin\left(\frac{\alpha_i+\alpha_{i-1}}{2}\right)\cos\left(\frac{\alpha_i-\alpha_{i-1}}{2}\right) & (-1)^{i+1}\cos\left(\frac{\alpha_{i-1}-\alpha_i}{2}\right)\cos\left(\frac{\alpha_{i-1}+\alpha_i}{2}\right) \end{bmatrix} \begin{bmatrix} \cos\theta_i \\ \sin\theta_i \end{bmatrix}$$

$$= 2r\begin{bmatrix} \sin\left(\frac{\alpha_{i-1}-\alpha_i}{2}\right) \\ (-1)^i\cos\left(\frac{\alpha_{i-1}-\alpha_i}{2}\right) \end{bmatrix} \begin{bmatrix} \sin\left(\frac{\alpha_{i-1}+\alpha_i}{2}\right) & -\cos\left(\frac{\alpha_{i-1}+\alpha_i}{2}\right) \end{bmatrix} \begin{bmatrix} \cos\theta_i \\ \sin\theta_i \end{bmatrix}$$

Using equation (15), we directly have: $\alpha_i = 2\theta_i - \alpha_{i-1}$. Thus, the above equation reduces to:

$$= 2r \begin{bmatrix} \sin(\theta_i - \alpha_i) \\ (-1)^i \cos(\theta_i - \alpha_i) \end{bmatrix} [\sin(\theta_i) \quad -\cos(\theta_i)] \begin{bmatrix} \cos\theta_i \\ \sin\theta_i \end{bmatrix} = \begin{bmatrix} 0. \\ 0. \end{bmatrix}$$

Hence, the linear functions given by equations (16) and (17) are equal at $\varphi = \theta_i$. This proves that the function is continuous for Case 1.

**Proof for Case 2**:
Using equation (14), we know that $\sigma_\Theta$ realizes different linear functions for $\theta_0 < \varphi < \theta_0 + \epsilon$ and $\theta_{2n} - \epsilon < \varphi < \theta_{2n}$.
Thus, for $\sigma_\Theta$ to be continuous, we require that the two linear functions be the same at the boundary i.e $\varphi = \theta_0$.
As before, we first write the input $(z_1, \ z_2)$ in terms of shifted polar coordinates i.e: $(r\cos(\varphi), \ r\sin(\varphi))$.
Thus, the function for $\theta_0 < \varphi < \theta_0 + \epsilon$ is given by:

$$\sigma_\Theta (r\cos\varphi, r\sin\varphi) = \begin{bmatrix} \cos\alpha_0 & \sin\alpha_0 \\ \sin\alpha_0 & -\cos\alpha_0 \end{bmatrix} \begin{bmatrix} r\cos\varphi \\ r\sin\varphi \end{bmatrix} \tag{18}$$

Similarly, the function for $\theta_{2n} - \epsilon < \phi < \theta_{2n}$ is given by:

$$\sigma_\Theta (r\cos\varphi, r\sin\varphi) = \begin{bmatrix} \cos\alpha_{2n-1} & \sin\alpha_{2n-1} \\ (-1)^i \sin\alpha_{2n-1} & (-1)^{i+1} \cos\alpha_{2n-1} \end{bmatrix} \begin{bmatrix} r\cos\varphi \\ r\sin\varphi \end{bmatrix} \tag{19}$$

Using equation (15), $\alpha_{2n-1}$ is given as follows:

$$\alpha_{2n-1} = 2 \sum_{i=0}^{2n-1} \theta_{2n-1-i}(-1)^i = 2 \sum_{i=0}^{n-1} (\theta_{2i+1} - \theta_{2i}) = 2\pi$$

Thus, equation (19) reduces to:

$$\sigma_\Theta (r\cos\varphi, r\sin\varphi) = \begin{bmatrix} 1 & 0 \\ 0 & 1 \end{bmatrix} \begin{bmatrix} r\cos\varphi \\ r\sin\varphi \end{bmatrix} \tag{20}$$

The difference in the function values at the boundary i.e $\varphi = \theta_0$, obtained by subtracting equations (20) and (18) is given as follows:

$$\begin{bmatrix} 1 & 0 \\ 0 & 1 \end{bmatrix} \begin{bmatrix} r\cos\theta_0 \\ r\sin\theta_0 \end{bmatrix} - \begin{bmatrix} \cos\alpha_0 & \sin\alpha_0 \\ \sin\alpha_0 & -\cos\alpha_0 \end{bmatrix} \begin{bmatrix} r\cos\theta_0 \\ r\sin\theta_0 \end{bmatrix}$$

$$= r \begin{bmatrix} 1 - \cos\alpha_0 & -\sin\alpha_0 \\ -\sin\alpha_0 & 1 + \cos\alpha_0 \end{bmatrix} \begin{bmatrix} \cos\theta_0 \\ \sin\theta_0 \end{bmatrix}$$

Using the trigonometric identities: $1 - \cos(\theta) = 2\sin^2(\theta/2)$, $1 + \cos(\theta) = 2\cos^2(\theta/2)$ and $\sin(\theta) = 2\sin(\theta/2)\cos(\theta/2)$, we have:

$$= r \begin{bmatrix} 2\sin^2(\frac{\alpha_0}{2}) & -2\sin(\frac{\alpha_0}{2})\cos(\frac{\alpha_0}{2}) \\ -2\sin(\frac{\alpha_0}{2})\cos(\frac{\alpha_0}{2}) & 2\cos^2(\frac{\alpha_0}{2}) \end{bmatrix} \begin{bmatrix} \cos\theta_0 \\ \sin\theta_0 \end{bmatrix}$$

$$= 2r \begin{bmatrix} \sin(\frac{\alpha_0}{2}) \\ -\cos(\frac{\alpha_0}{2}) \end{bmatrix} [\sin(\frac{\alpha_0}{2}) \quad -\cos(\frac{\alpha_0}{2})] \begin{bmatrix} \cos\theta_0 \\ \sin\theta_0 \end{bmatrix}$$

Using equation (15), we have: $\alpha_0 = 2\theta_0$. Thus, the above equation reduces to:

$$= 2r \begin{bmatrix} \sin(\theta_0) \\ -\cos(\theta_0) \end{bmatrix} [\sin(\theta_0) \quad -\cos(\theta_0)] \begin{bmatrix} \cos\theta_0 \\ \sin\theta_0 \end{bmatrix} = \begin{bmatrix} 0. \\ 0. \end{bmatrix}$$

Hence, the linear functions given by equations (18) and (20) are equal at $\varphi = \theta_0$.
This proves that the function is continuous for Case 2. □

## B    SELECTION OF $\gamma$ USING CROSS VALIDATION

Using 5000 held out samples from CIFAR-10, we tested 6 different values of $\gamma$ shown in Table 3 and selected $\gamma = 0.5$ because it resulted in less than $0.5\%$ decrease in standard accuracy while $4.96\%$ increase in provably robust accuracy. We used the LipConvnet-5 network with the $2D$ Householder activation function of order 2 i.e $\sigma_\theta$.

| Architecture | $\gamma$ | Standard Accuracy | Provable Robust Acc. ($\rho =$) | | | Increase (108/255) |
|---|---|---|---|---|---|---|
| | | | 36/255 | 72/255 | 108/255 | |
| | 0. | 75.82% | 59.66% | 42.78% | 26.92% | _ |
| | 0.10 | 75.58% | 59.74% | 42.94% | 28.04% | +0.94% |
| | 0.25 | 75.54% | 60.22% | 43.98% | 29.50% | +2.58% |
| LipConvnet-5 | 0.50 | 75.30% | **60.08%** | **45.36%** | **31.88%** | **+4.96%** |
| | 0.75 | 74.14% | 60.36% | 46.08% | 33.44% | +6.52% |
| | 1.00 | 73.86% | 60.30% | 46.80% | 34.60% | +7.68% |

Table 3: Results for provable robustness against adversarial examples on the CIFAR-10 dataset for cross validation using 5000 held out samples from the training set.

## C    DIFFERENCES BETWEEN $l_1$ AND $l_2$ CERTIFICATE

By the equivalence of norms, we have the following result:

$$\mathbf{x} \in \mathbb{R}^d, \qquad \frac{1}{\sqrt{d}}\|\mathbf{x}\|_1 \leq \|\mathbf{x}\|_2 \leq \sqrt{d}\|\mathbf{x}\|_1 \tag{21}$$

Let us assume that we have an $l_1$ certificate for the input $\mathbf{x}$ so that the prediction of a neural network remains constant in a region of $l_1$ radius $\rho_1$ around the input $\mathbf{x}$ i.e:

$$\|\mathbf{x}^* - \mathbf{x}\|_1 \leq \rho_1 \tag{22}$$

We want to compute the $l_2$ certificate implied by the certificate given in equation (22). Let $\rho_2$ be the $l_2$ certificate so that:

$$\|\mathbf{x}^* - \mathbf{x}\|_2 \leq \rho_2 \implies \|\mathbf{x}^* - \mathbf{x}\|_1 \leq \rho_1$$

Using equation (21), we have the following set of equations:

$$\|\mathbf{x}^* - \mathbf{x}\|_2 \leq \rho_2 \implies \frac{1}{\sqrt{d}}\|\mathbf{x}^* - \mathbf{x}\|_1 \leq \rho_2 \implies \|\mathbf{x}^* - \mathbf{x}\|_1 \leq \sqrt{d}\rho_2$$

Using equation (22), we have the following:

$$\sqrt{d}\rho_2 \leq \rho_1 \implies \rho_2 \leq \frac{\rho_1}{\sqrt{d}}$$

Hence, the $l_2$ norm certificate induced by the $l_1$ norm certificate can be significantly smaller for high dimensional inputs. Using the $l_1$ certificate used in Levine & Feizi (2021) i.e $\rho_1 = 4$ and $d = 32\sqrt{3}$ for CIFAR-10 and CIFAR-100, we get an implied certificate of $\rho_2 = 4/(32\sqrt{3}) = 0.07225$. In contrast, in this work we study $l_2$ robustness for much higher certificate values i.e $36/255 = 0.14118$.

## D    RESULTS USING REVISED LIPSCHITZ CONSTANTS

In Tables 4 and 5, we show results where the Lipschitz constant was computed using product of Lipschitz constant of all layers. The Lipschitz constant of each layer was computed using direct power iteration on the linear layer (using 50 iterations) and not using the approximation bound provided in Singla & Feizi (2021).

| Architecture | Methods | Standard Accuracy | Provable Robust Acc. ($\rho =$) | | | Increase |
| --- | --- | --- | --- | --- | --- | --- |
| | | | 36/255 | 72/255 | 108/255 | (Standard) |
| LipConvnet-5 | SOC + MaxMin | 42.71% | 27.86% | 17.45% | 9.99% | _ |
| | **+ LLN** | 45.86% | 31.93% | 21.17% | 13.21% | +3.15% |
| | **+ HH** | **46.36%** | 32.64% | 21.19% | 13.12% | **+3.65%** |
| | **+ CR** | 45.82% | **32.99%** | **22.48%** | 14.79% | +3.11% |
| LipConvnet-10 | SOC + MaxMin | 43.72% | 29.39% | 18.56% | 11.16% | _ |
| | **+ LLN** | 46.88% | 33.32% | 22.08% | 13.87% | +3.16% |
| | **+ HH** | **47.96%** | 34.30% | 22.35% | 14.48% | **+4.24%** |
| | **+ CR** | 47.07% | **34.53%** | **23.50%** | **15.66%** | +3.35% |
| LipConvnet-15 | SOC + MaxMin | 42.92% | 28.81% | 17.93% | 10.73% | _ |
| | **+ LLN** | 47.72% | 33.52% | 21.89% | 13.76% | +4.80% |
| | **+ HH** | **47.72%** | 33.97% | 22.45% | 13.81% | **+4.80%** |
| | **+ CR** | 47.61% | **34.54%** | **23.16%** | **15.09%** | +4.69% |
| LipConvnet-20 | SOC + MaxMin | 43.06% | 29.34% | 18.66% | 11.20% | _ |
| | **+ LLN** | 46.86% | 33.48% | 22.14% | 14.10% | +3.80% |
| | **+ HH** | 47.71% | 34.22% | 22.93% | 14.57% | +4.65% |
| | **+ CR** | **47.84%** | **34.77%** | **23.70%** | **15.83%** | **+4.78%** |
| LipConvnet-25 | SOC + MaxMin | 43.37% | 28.59% | 18.17% | 10.85% | _ |
| | **+ LLN** | 46.32% | 32.87% | 21.53% | 13.86% | +2.95% |
| | **+ HH** | **47.70%** | 34.00% | 22.67% | 14.57% | **+4.33%** |
| | **+ CR** | 46.87% | **34.09%** | **23.41%** | **15.61%** | +3.50% |
| LipConvnet-30 | SOC + MaxMin | 42.87% | 28.74% | 18.47% | 11.20% | _ |
| | **+ LLN** | 46.18% | 32.82% | 21.52% | 13.52% | +3.31% |
| | **+ HH** | 46.80% | 33.72% | 22.70% | 14.31% | +3.93% |
| | **+ CR** | **46.92%** | **34.17%** | **23.21%** | **15.84%** | **+4.05%** |
| LipConvnet-35 | SOC + MaxMin | 42.42% | 28.34% | 18.10% | 10.96% | _ |
| | **+ LLN** | 45.22% | 32.10% | 21.28% | 13.25% | +2.80% |
| | **+ HH** | 46.21% | 32.80% | 21.55% | 14.13% | +3.79% |
| | **+ CR** | **46.88%** | **33.64%** | **23.34%** | **15.73%** | **+4.46%** |
| LipConvnet-40 | SOC + MaxMin | 41.84% | 28.00% | 17.40% | 10.28% | _ |
| | **+ LLN** | 42.94% | 29.71% | 19.30% | 11.99% | +1.10% |
| | **+ HH** | **45.84%** | **32.79%** | 21.98% | 14.07% | **+4.00%** |
| | **+ CR** | 45.03% | 32.56% | **22.37%** | **14.76%** | +3.19% |

Table 4: Results for provable robustness against adversarial examples on the CIFAR-100 dataset. For all networks in this table, the maximum deviation of the Lipschitz constant from 1 was measured to be $2.4609 \times 10^{-5}$. The values shown in red were reduced by $0.01\%$ from the corresponding values in Table 1 due to the correction in Lipschitz constant. All other values remained unchanged.

| Architecture | Methods | Standard Accuracy | Provable Robust Acc. ($\rho =$) | | | Increase |
|---|---|---|---|---|---|---|
| | | | 36/255 | 72/255 | 108/255 | (108/255) |
| LipConvnet-5 | SOC + MaxMin | 75.78% | 59.18% | 42.01% | 27.09% | _ |
| | **+ HH** | **76.30%** | 60.12% | 43.20% | 27.39% | +0.30% |
| | **+ CR** | 75.31% | **60.37%** | **45.62%** | **32.38%** | **+5.29%** |
| LipConvnet-10 | SOC + MaxMin | 76.45% | 60.86% | 44.15% | 29.15% | _ |
| | **+ HH** | **76.86%** | 61.52% | 44.91% | 29.90% | +0.75% |
| | **+ CR** | 76.23% | **62.57%** | **47.70%** | **34.15%** | **+5.00%** |
| LipConvnet-15 | SOC + MaxMin | 76.68% | 61.36% | 44.27% | 29.66% | _ |
| | **+ HH** | **77.41%** | 61.92% | 45.60% | 31.10% | +1.44% |
| | **+ CR** | 76.39% | **62.96%** | **48.47%** | **35.47%** | **+5.81%** |
| LipConvnet-20 | SOC + MaxMin | 76.90% | 61.87% | 45.79% | 31.08% | _ |
| | **+ HH** | **76.99%** | 61.76% | 45.59% | 30.99% | -0.09% |
| | **+ CR** | 76.34% | **62.63%** | **48.68%** | **36.04%** | **+4.96%** |
| LipConvnet-25 | SOC + MaxMin | 75.24% | 60.17% | 43.54% | 28.60% | _ |
| | **+ HH** | **76.37%** | 61.50% | 44.72% | 29.83% | +1.23% |
| | **+ CR** | 75.21% | **61.98%** | **47.93%** | **34.92%** | **+6.32%** |
| LipConvnet-30 | SOC + MaxMin | 74.51% | 59.06% | 42.46% | 28.04% | _ |
| | **+ HH** | **75.25%** | 59.90% | 43.85% | 29.35% | +1.30% |
| | **+ CR** | 74.23% | **60.64%** | **46.51%** | **34.08%** | **+6.04%** |
| LipConvnet-35 | SOC + MaxMin | 73.73% | 58.50% | 41.75% | 27.20% | _ |
| | **+ HH** | **75.44%** | 61.05% | 45.38% | 30.85% | +3.65% |
| | **+ CR** | 74.25% | **61.30%** | **47.60%** | **35.21%** | **+8.01%** |
| LipConvnet-40 | SOC + MaxMin | 71.63% | 54.39% | 37.92% | 24.13% | _ |
| | **+ HH** | **73.24%** | 58.12% | 42.23% | 28.48% | +4.35% |
| | **+ CR** | 72.59% | **59.04%** | **44.92%** | **32.87%** | **+8.74%** |

Table 5: Results for provable robustness against adversarial examples on the CIFAR-10 dataset. For all networks in this table, the maximum deviation of the Lipschitz constant from $1$ was measured to be $2.2072 \times 10^{-5}$. The values shown in red were reduced by $0.01\%$ from the corresponding values in Table 2 due to the correction in Lipschitz constant. All other values remained unchanged.

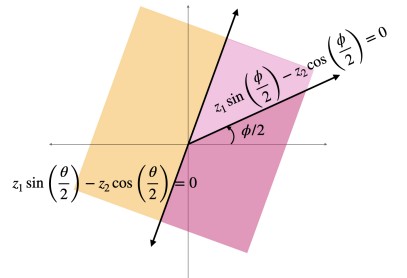 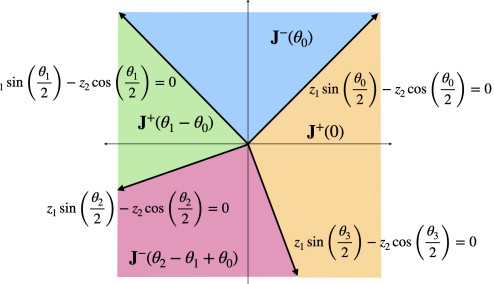

(a) The output of $\sigma_\theta$ always lies in the pink region. Applying $\sigma_\phi$ on this where $\phi \neq \theta + 2n\pi, n \in \mathbb{Z}$ further divides this region into two linear regions (light and original pink).

(b) In each colored region, the function is linear with the Jacobian mentioned. For the function to be continuous, we must have $\theta_3 - \theta_2 + \theta_1 - \theta_0 = \pi$. Thus, any 3 of $\{\theta_0, \theta_1, \theta_2, \theta_3\}$ can be chosen as learnable parameters.

Figure 2: Constructing Higher Order Householder activations ($\mathbf{J}^+$ and $\mathbf{J}^-$ defined in equation (2))

## E  VERIFICATION THAT $\sigma_\theta(z_1, z_2)$ ALWAYS LIES ON ONE SIDE OF THE HYPERPLANE

Consider the case: $z_1 \sin(\theta/2) - z_2 \cos(\theta/2) > 0$
In this case $\sigma_\theta(z_1, z_2) = (z_1, z_2)$ and the result follows directly.
Consider the other case: $z_1 \sin(\theta/2) - z_2 \cos(\theta/2) \leq 0$

$$\begin{bmatrix} a_1 \\ a_2 \end{bmatrix} = \begin{bmatrix} \cos\theta & \sin\theta \\ \sin\theta & -\cos\theta \end{bmatrix} \begin{bmatrix} z_1 \\ z_2 \end{bmatrix} = \begin{bmatrix} z_1 \cos\theta + z_2 \sin\theta \\ z_1 \sin\theta - z_2 \cos\theta \end{bmatrix}$$

$$a_1 \sin(\theta/2) - a_2 \cos(\theta/2) = (z_1 \cos\theta + z_2 \sin\theta) \sin(\theta/2) - (z_1 \sin\theta - z_2 \cos\theta) \cos(\theta/2)$$
$$= z_1 (\cos\theta \sin(\theta/2) - \sin\theta \cos(\theta/2)) + z_2 (\sin\theta \sin(\theta/2) + \cos\theta \cos(\theta/2))$$
$$= -z_1 \sin(\theta/2) + z_2 \cos(\theta/2)$$

Since $z_1 \sin(\theta/2) - z_2 \cos(\theta/2) \leq 0$, we have $-z_1 \sin(\theta/2) + z_2 \cos(\theta/2) \geq 0$.

## F  HIGHER ORDER HOUSEHOLDER ACTIVATION FUNCTIONS

We know that $\mathrm{MaxMin}(z_1, z_2) = (\max(z_1, z_2), \min(z_1, z_2))$ where $z_1, z_2 \in \mathbb{R}$. Because $\max(z_1, z_2) > \min(z_1, z_2)$, applying MaxMin again gives the same result i.e $\mathrm{MaxMin} \circ \mathrm{MaxMin} = \mathrm{MaxMin}$. Now consider the function $\sigma_\theta$ (discussed in the maintext, given below for convenience):

$$\sigma_\theta(z_1, z_2) = \begin{cases} \begin{bmatrix} 1 & 0 \\ 0 & 1 \end{bmatrix} \begin{bmatrix} z_1 \\ z_2 \end{bmatrix} & \text{if } z_1 \sin(\theta/2) - z_2 \cos(\theta/2) > 0 \\ \begin{bmatrix} \cos\theta & \sin\theta \\ \sin\theta & -\cos\theta \end{bmatrix} \begin{bmatrix} z_1 \\ z_2 \end{bmatrix} & \text{if } z_1 \sin(\theta/2) - z_2 \cos(\theta/2) \leq 0 \end{cases}$$

From Figure 1, we observe that if $(u_1, u_2) = \sigma_\theta(z_1, z_2)$, then $(u_1, u_2)$ always lies on the right side of the hyperplane (pink colored region). In other words, $u_1 \sin(\theta/2) - u_2 \cos(\theta/2) > 0 \ \forall z_1, z_2$ (proof in Appendix E). This further implies that $\sigma_\theta \circ \sigma_\theta = \sigma_\theta$.

However from Figure 2a in the maintext, we observe that if we use a different angle $\phi$ where $\phi \neq \theta + 2n\pi$ for some $n \in \mathbb{Z}$, then $\sigma_\phi(u_1, u_2) \neq (u_1, u_2)$ for all $(u_1, u_2)$ in the pink colored region ($u_1 \sin(\theta/2) - u_2 \cos(\theta/2) > 0$). This motivates us to construct the function $\sigma^{(n)} : \mathbb{R}^2 \to \mathbb{R}^2$ defined as follows:

$$\sigma^{(n)} = \underbrace{\sigma_\theta \circ \sigma_\theta \circ \sigma_\theta \dots \circ \sigma_\theta}_{n \text{ times, } \theta\text{'s can be different}} \tag{23}$$

Clearly, $\sigma^{(n)}$ has a larger number of linear regions than $\sigma_\theta$ and thus expected to have more expressive power. However, a drawback of using $\sigma^{(n)}$ is that it requires a sequential application of $\sigma_\theta$ which can

be expensive if the number of iterations $n$ is large. To address this limitation, first observe that $\sigma_\theta$ realizes the same linear function for $(z_1, z_2)$ and $(cz_1, cz_2)$ when $c > 0$ i.e $\nabla_{(z_1, z_2)} \sigma_\theta = \nabla_{(cz_1, cz_2)} \sigma_\theta$. Since $\sigma_\theta$ is piecewise linear, $\sigma_\theta(cz_1, cz_2) = c\sigma_\theta(z_1, z_2)$. Thus, the input of the next function in the iteration is scaled by $c$ as well and its linear function (or the Jacobian) remains unchanged. By induction, same holds for all the subsequent iterations. By chain rule, the Jacobian of composition of functions is equal to the product of Jacobian of each individual function. Since the Jacobian of each function is unchanged on scaling by $c > 0$, the Jacobian $\nabla_{(z_1, z_2)} \sigma^{(n)}$ also remains unchanged: $\nabla_{(z_1, z_2)} \sigma^{(n)} = \nabla_{(cz_1, cz_2)} \sigma^{(n)}$. This suggests that it is possible to determine the Jacobian $\nabla_{(z_1, z_2)} \sigma^{(n)}$ for the input $(z_1, z_2)$ by first converting to the polar coordinates $(\sqrt{z_1^2 + z_2^2}, \varphi)$ and then using the phase angle $\varphi$ alone (where $\cos(\varphi) = z_1/\sqrt{z_1^2 + z_2^2}$, $\sin(\varphi) = z_2/\sqrt{z_1^2 + z_2^2}$).

This motivates us to construct another GNP piecewise linear activation that only depends on the phase of the input but unlike $\sigma_\theta$, it is allowed to have more than 2 linear regions without requiring a sequential application. This construction is given in the following theorem (example in Figure 2b):

**Theorem 2.** *Given:* $0 \leq \theta_0 < \theta_1 \cdots < \theta_{2n} = 2\pi + \theta_0$ *such that* $\sum_{i=0}^{n-1}(\theta_{2i+1} - \theta_{2i}) = \pi$ *and* $\alpha_i = 2\sum_{j=0}^{i} \theta_{i-j}(-1)^j$, *The function* $\sigma_\Theta : \mathbb{R}^2 \to \mathbb{R}^2$ *is continuous, GNP and 1-Lipschitz where* $\Theta = [\theta_0, \theta_1, \ldots, \theta_{2n-1}]$ *(also called Householder Activation of order $n$ in 2 dimensions):*

$$\sigma_\Theta(z_1, z_2) = \begin{bmatrix} \cos\alpha_i & \sin\alpha_i \\ (-1)^i \sin\alpha_i & (-1)^{i+1}\cos\alpha_i \end{bmatrix} \begin{bmatrix} z_1 \\ z_2 \end{bmatrix} \qquad \theta_i \leq \varphi < \theta_{i+1}$$

*where* $\varphi \in [\theta_0, \ \theta_{2n} = 2\pi + \theta_0)$ *and* $\cos(\varphi) = z_1/\sqrt{z_1^2 + z_2^2}$, $\sin(\varphi) = z_2/\sqrt{z_1^2 + z_2^2}$.

Using the definition of $\alpha_i$, $\alpha_{2n-1}$ can be computed as follows:

$$\alpha_{2n-1} = 2\sum_{j=0}^{2n-1} \theta_{2n-1-j}(-1)^j = 2\sum_{j=0}^{n-1}(\theta_{2n-1-2j} - \theta_{2n-2-2j}) = 2\sum_{j=0}^{n-1}(\theta_{2j+1} - \theta_{2j}) = 2\pi$$

$$\sigma_\Theta(z_1, z_2) = \begin{bmatrix} \cos\alpha_{2n-1} & \sin\alpha_{2n-1} \\ -\sin\alpha_{2n-1} & \cos\alpha_{2n-1} \end{bmatrix} \begin{bmatrix} z_1 \\ z_2 \end{bmatrix} = (z_1, z_2) \qquad \theta_{2n-1} \leq \varphi < \theta_{2n}(=\theta_0 + 2\pi)$$

By continuity, $\sigma_\Theta(z_1, z_2) = (z_1, z_2)$ for $\varphi = \theta_0$. Thus if we set $\theta_0 = 0$, $\sigma_\Theta$ is fixed to be identity when $\varphi = 0$ (or $z_2 = 0$). However, a learnable $\theta_0$ offers the flexibility of choosing arbitrary intervals around the $\varphi = \theta_0$ to be the identity function (while of course allowing $\theta_0 = 0$). Since we can choose any interval of $2\pi$ for the phase angle, we choose $\varphi \in [\theta_0, \ \theta_{2n} = \theta_0 + 2\pi)$ instead of the usual $[0, \ 2\pi)$ to allow this possibility. We call $(\sqrt{z_1^2 + z_2^2}, \varphi)$, the *shifted polar coordinates*.

Additionally, we make the following observations about Theorem 2. First, by construction $\sigma_\Theta$ has $2n$ linear regions. Second, since $\sum_{i=0}^{n-1}(\theta_{2i+1} - \theta_{2i}) = \pi$, the sum of angles subtended by linear regions with determinant $-1$ equals $\pi$. This in turn implies that sum of angles subtended by linear regions with determinant $+1$ must also equal $\pi$. Third, again using $\sum_{j=0}^{n-1}(\theta_{2j+1} - \theta_{2j}) = \pi$, we know that only $2n-1$ of the $2n$ parameters in $[\theta_0, \ \theta_1, \ \theta_2, \ \ldots, \ \theta_{2n-1}]$ can be chosen independently implying that $\sigma_\Theta$ has $2n-1$ learnable parameters. In contrast, $\sigma^{(n)}$ has only $n$ learnable parameters. Fourth, when $n = 1$, $\sigma_\Theta$ reduces to the original $\sigma_\theta$ activation.

Because every function of the form $\sigma^{(n)}$ (equation (23)) can have potentially $2^n$ linear regions, while $\sigma_\Theta$ has only $2n$ linear regions, $\sigma_\Theta$ cannot express every function of the form $\sigma^{(n)}$. The primary benefit of using $\sigma_\Theta$ is that it can be easily applied by first determining the angle $\varphi$ (using shifted polar coordinates), the region $[\theta_i, \theta_{i+1})$ to which $\varphi$ belongs and the Jacobian for this region. This requires 1 multiplication with the Jacobian instead of $n$ required for $\sigma^{(n)}$.

## G    EXTENSION TO HIGHER DIMENSIONS

We introduced Householder activation function of Order 1 in $m$ dimensions in maintext Definition 1. However, it suffers from the limitation that it has only 2 linear regions thus limiting its expressive power. The construction given in Appendix Section F allows more than 2 linear regions but is valid only for 2 dimensional inputs. This motivates us to construct Householder activations that depend on all the $m$ components of input $\mathbf{z} \in \mathbb{R}^m$, $(m \geq 3)$ while allowing for more than 2 linear regions.

A straightforward way of constructing such an activation function is to apply an orthogonal matrix $\mathbf{Q} \in \mathbb{R}^{m \times m}$, followed by dividing the output $\mathbf{Qz}$ into groups of size 2 each and then applying $\sigma_\theta$ to each group. However, since 1-Lipschitz neural networks involve multiplication with an orthogonal weight matrix followed by GNP activation anyway, this construction is trivial because it does not lead to additional gains in expressive power over using 2 dimensional $\sigma_\Theta$ activation functions.

Recall that the function $\sigma_\mathbf{v}$ is given by the following equation:

$$\sigma_\mathbf{v}(\mathbf{z}) = \begin{cases} \mathbf{z}, & \mathbf{v}^T\mathbf{z} > 0, \\ (\mathbf{I} - 2\mathbf{v}\mathbf{v}^T)\mathbf{z}, & \mathbf{v}^T\mathbf{z} \leq 0. \end{cases} \tag{24}$$

By a similar analysis as for the 2-dimensional case (Figure 2 in maintext), a repeated application of $\sigma_\mathbf{v}$ leads to increased number of linear regions and thus higher expressive power. This motivates us to construct the function $\sigma^{(m,n)} : \mathbb{R}^m \to \mathbb{R}^m$ by applying the function $\sigma_\mathbf{v}$ (equation (24)) $n$ times iteratively with different learnable parameter $\mathbf{v}$ at each iteration:

$$\sigma^{(m,n)} = \underbrace{\sigma_\mathbf{v} \circ \sigma_\mathbf{v} \circ \sigma_\mathbf{v} \ldots \circ \sigma_\mathbf{v}}_{n \text{ times, } \mathbf{v}\text{'s can be different}}$$

Since $\sigma_\mathbf{v}$ realizes the same linear function for both the inputs $\mathbf{z}$ and $c\mathbf{z}$ i.e $\nabla_\mathbf{z}\, \sigma_\mathbf{v} = \nabla_{c\mathbf{z}}\, \sigma_\mathbf{v}$ when $c > 0$, $\sigma^{(m,n)}$ satisfies this property as well. This suggests that it is possible to determine the Jacobian of $\sigma^{(m,n)}$ for the given input $\mathbf{z}$ by projecting $\mathbf{z}$ onto a unit sphere: $\mathbf{z}/\|\mathbf{z}\|_2$. Moreover, we want our constructed function to have at least $2n$ linear regions while requiring $k$ iterations of $\sigma_\mathbf{v}$ where $k$ is independent of $n$. This motivates the following open question:

**Open Problem.** *Can non-trivial order-$n$ ($n > 1$) householder activation functions with $2n$ linear regions be constructed for $m$ dimensional input ($m > 2$) using $k$ iterations of $\sigma_\mathbf{v}$ where $k$ is independent of $n$ (but may depend on $m$)?*

## H    Additional results on CIFAR-10

The rows "BCOP", "Cayley" and "SOC (baseline)" all use the MaxMin activation function. "SOC + HH" replaces MaxMin with 2D Householder activation of order 1 ($\sigma_\theta$), "+ CR" adds Certificate Regularization (CR) with $\gamma = 0.1$ (while using $\sigma_\theta$ as the activation function).

In Table 9, the row "SOC + HH$^{(2)}$" uses Householder activation of order 2 ($\sigma_\Theta$ defined in Theorem 2), "+ CR" adds Certificate Regularization (CR) with $\gamma = 0.1$ (while using the HH activation of order 2 i.e $\sigma_\Theta$ as the activation function).

None of the results in Tables 7, 8 and 9 include Last Layer Normalization (LLN).

| Architecture | Running times (seconds) | |
| --- | --- | --- |
| | MaxMin | HH |
| LipConvnet-5 | 3.7864 | 3.86 |
| LipConvnet-10 | 5.3677 | 5.6014 |
| LipConvnet-15 | 7.234 | 7.3503 |
| LipConvnet-20 | 9.536 | 9.3753 |
| LipConvnet-25 | 11.0512 | 11.2692 |
| LipConvnet-30 | 12.5135 | 13.6866 |
| LipConvnet-35 | 14.5539 | 15.0921 |
| LipConvnet-40 | 17.1907 | 17.1928 |

Table 6: Inference times for various networks on the complete test dataset of CIFAR-10 with $10000$ samples. None of these networks include Last Layer Normalization (LLN).

| Architecture | Methods | Standard Accuracy | Provable Robust Acc. ($\rho =$) | | | Increase |
| --- | --- | --- | --- | --- | --- | --- |
| | | | 36/255 | 72/255 | 108/255 | (108/255) |
| LipConvnet-5 | BCOP | 74.25% | 58.01% | 40.34% | 25.21% | -1.88% |
| | Cayley | 72.37% | 55.92% | 38.65% | 24.27% | -2.82% |
| | SOC (baseline) | 75.78% | 59.18% | 42.01% | 27.09% | (+0%) |
| | **SOC + HH** | **76.30%** | 60.12% | 43.20% | 27.39% | +0.30% |
| | **+ CR** | 75.31% | **60.37%** | **45.62%** | **32.38%** | **+5.29%** |
| LipConvnet-10 | BCOP | 74.47% | 58.48% | 40.77% | 26.16% | -2.99% |
| | Cayley | 74.30% | 57.99% | 40.75% | 25.93% | -3.22% |
| | SOC (baseline) | 76.45% | 60.86% | 44.15% | 29.15% | (+0%) |
| | **SOC + HH** | **76.86%** | 61.52% | 44.91% | 29.90% | +0.75% |
| | **+ CR** | 76.23% | **62.57%** | **47.70%** | **34.15%** | **+5.00%** |
| LipConvnet-15 | BCOP | 73.86% | 57.39% | 39.33% | 24.86% | -4.80% |
| | Cayley | 71.92% | 54.55% | 37.67% | 23.50% | -6.16% |
| | SOC (baseline) | 76.68% | 61.36% | 44.28% | 29.66% | (+0%) |
| | **SOC + HH** | **77.41%** | 61.92% | 45.60% | 31.10% | +1.44% |
| | **+ CR** | 76.39% | **62.96%** | **48.47%** | **35.47%** | **+5.81%** |
| LipConvnet-20 | BCOP | 69.84% | 52.10% | 34.75% | 21.09% | -9.99% |
| | Cayley | 68.87% | 51.88% | 35.56% | 21.72% | -9.36% |
| | SOC (baseline) | **76.90%** | 61.87% | 45.79% | 31.08% | (+0%) |
| | **SOC + HH** | 76.99% | 61.76% | 45.59% | 30.99% | -0.09% |
| | **+ CR** | 76.34% | **62.63%** | **48.69%** | **36.04%** | **+4.96%** |

Table 7: Results for provable robustness on the CIFAR-10 dataset using shallow networks. None of these results include Last Layer Normalization (LLN).

| Architecture | Methods | Standard Accuracy | Provable Robust Acc. ($\rho =$) | | | Increase |
| --- | --- | --- | --- | --- | --- | --- |
| | | | 36/255 | 72/255 | 108/255 | (108/255) |
| LipConvnet-25 | BCOP | 68.47% | 49.92% | 31.99% | 18.62% | -9.98% |
| | Cayley | 64.00% | 45.55% | 29.24% | 16.99% | -11.61% |
| | SOC (baseline) | 75.24% | 60.17% | 43.55% | 28.60% | (+0%) |
| | **SOC + HH** | **76.37%** | 61.50% | 44.72% | 29.83% | +1.23% |
| | **+ CR** | 75.21% | **61.98%** | **47.93%** | **34.92%** | **+6.32%** |
| LipConvnet-30 | BCOP | 64.11% | 43.39% | 25.02% | 12.15% | -15.90% |
| | Cayley | 58.83% | 38.68% | 22.07% | 10.68% | -17.37% |
| | SOC (baseline) | 74.51% | 59.06% | 42.46% | 28.05% | (+0%) |
| | **SOC + HH** | **75.25%** | 59.90% | 43.85% | 29.35% | +1.30% |
| | **+ CR** | 74.23% | **60.64%** | **46.51%** | **34.08%** | **+6.03%** |
| LipConvnet-35 | BCOP | 63.05% | 41.71% | 23.30% | 11.36% | -15.84% |
| | Cayley | 53.55% | 32.37% | 16.18% | 6.33% | -20.87% |
| | SOC (baseline) | 73.73% | 58.50% | 41.75% | 27.20% | (+0%) |
| | **SOC + HH** | **75.44%** | 61.05% | 45.38% | 30.85% | +3.65% |
| | **+ CR** | 74.25% | **61.30%** | **47.60%** | **35.21%** | **+8.01%** |
| LipConvnet-40 | BCOP | 60.17% | 38.86% | 21.20% | 9.08% | -15.05% |
| | Cayley | 51.26% | 27.90% | 12.06% | 3.81% | -20.32% |
| | SOC (baseline) | 71.63% | 54.39% | 37.92% | 24.13% | (+0%) |
| | **SOC + HH** | **73.24%** | 58.12% | 42.24% | 28.48% | +4.35% |
| | **+ CR** | 72.59% | **59.04%** | **44.92%** | **32.87%** | **+8.74%** |

Table 8: Results for provable robustness against adversarial examples on the CIFAR-10 dataset. None of these results include Last Layer Normalization (LLN).

| Architecture | Methods | Standard Accuracy | Provable Robust Acc. ($\rho =$) | | | Increase |
| | | | 36/255 | 72/255 | 108/255 | (108/255) |
|---|---|---|---|---|---|---|
| LipConvnet-5 | SOC + HH$^{(2)}$ | 75.85% | 59.66% | 42.68% | 27.44% | +0.35% |
| | + CR | 74.85% | 60.56% | 44.96% | 31.98% | +4.59% |
| LipConvnet-10 | SOC + HH$^{(2)}$ | 76.80% | 61.44% | 44.91% | 29.70% | +0.55% |
| | + CR | 76.30% | 62.11% | 47.85% | 34.49% | +5.34% |
| LipConvnet-15 | SOC + HH$^{(2)}$ | 77.41% | 62.21% | 45.11% | 30.49% | +0.83% |
| | + CR | 75.73% | 62.21% | 47.92% | 35.26% | +5.60% |
| LipConvnet-20 | SOC + HH$^{(2)}$ | 76.69% | 61.58% | 45.39% | 30.89% | -0.19% |
| | + CR | 75.72% | 62.61% | 48.30% | 35.29% | +4.21% |
| LipConvnet-25 | SOC + HH$^{(2)}$ | 76.12% | 61.24% | 44.81% | 29.63% | +1.03% |
| | + CR | 75.38% | 61.94% | 47.67% | 34.22% | +5.62% |
| LipConvnet-30 | SOC + HH$^{(2)}$ | 75.09% | 60.01% | 44.22% | 29.39% | +1.34% |
| | + CR | 74.88% | 61.23% | 46.63% | 34.02% | +5.97% |
| LipConvnet-35 | SOC + HH$^{(2)}$ | 73.93% | 58.61% | 42.29% | 28.47% | +1.27% |
| | + CR | 74.14% | 60.72% | 46.67% | 34.64% | +7.44% |
| LipConvnet-40 | SOC + HH$^{(2)}$ | 70.90% | 54.96% | 38.94% | 24.90% | +0.77% |
| | + CR | 72.28% | 57.67% | 43.00% | 30.66% | +6.53% |

Table 9: Results for provable robustness on CIFAR-10 using HH activation of Order 2 ($\sigma_\Theta$). Increase (108/255) is calculated with respect to SOC baseline (from Tables 7, 8). None of these results include Last Layer Normalization (LLN).

| Architecture | Methods | Standard Accuracy | Provable Robust Acc. ($\rho =$) | | | Increase |
| | | | (36/255) | (72/255) | (108/255) | (Standard) |
|---|---|---|---|---|---|---|
| LipConvnet-5 | SOC (no LLN) | 75.78% | 59.18% | 42.01% | 27.09% | (+0%) |
| | **SOC + LLN** | **75.78%** | **59.58%** | **42.45%** | **27.20%** | **+0.00%** |
| LipConvnet-10 | SOC (no LLN) | 76.45% | 60.86% | 44.15% | 29.15% | (+0%) |
| | **SOC + LLN** | **76.69%** | **61.08%** | **44.04%** | **29.19%** | **+0.24%** |
| LipConvnet-15 | SOC (no LLN) | 76.68% | 61.36% | 44.28% | 29.66% | (+0%) |
| | **SOC + LLN** | **76.84%** | **61.94%** | **45.51%** | **30.28%** | **+0.16%** |
| LipConvnet-20 | SOC (no LLN) | **77.05%** | **61.87%** | **45.79%** | **31.08%** | **(+0%)** |
| | **SOC + LLN** | 76.71% | 61.44% | 44.92% | 30.19% | -0.34% |
| LipConvnet-25 | SOC (no LLN) | 75.24% | 60.17% | 43.55% | 28.60% | (+0%) |
| | **SOC + LLN** | **76.54%** | **61.21%** | **44.18%** | **29.47%** | **+1.30%** |
| LipConvnet-30 | SOC (no LLN) | **74.51%** | **59.06%** | **42.46%** | **28.05%** | **(+0%)** |
| | **SOC + LLN** | 74.26% | 58.97% | 41.82% | 26.93% | -0.25% |
| LipConvnet-35 | SOC (no LLN) | 73.73% | 58.50% | 41.75% | 27.20% | (+0%) |
| | **SOC + LLN** | **74.32%** | **59.05%** | **42.34%** | **28.14%** | **+0.59%** |
| LipConvnet-40 | SOC (no LLN) | 71.63% | 54.39% | 37.92% | 24.13% | (+0%) |
| | **SOC + LLN** | **74.03%** | **58.27%** | **41.75%** | **27.12%** | **+2.40%** |

Table 10: Results for provable robustness on the CIFAR-10 dataset with and without LLN

# I  ADDITIONAL RESULTS ON CIFAR-100

All results in Tables 13, 14 and 15 include Last Layer Normalization (LLN).

The rows "BCOP", "Cayley" and "SOC (baseline)" all use the $\mathrm{MaxMin}$ activation function. "SOC + HH" replaces $\mathrm{MaxMin}$ with 2D Householder activation of order 1 ($\sigma_\theta$), "+ CR" adds Certificate Regularization (CR) with $\gamma = 0.1$ (while using $\sigma_\theta$ as the activation function).

In Table 15, the row "SOC + HH$^{(2)}$" uses Householder activation of order 2 ($\sigma_\Theta$ defined in Theorem 2), "+ CR" adds Certificate Regularization (CR) with $\gamma = 0.1$ (while using the HH activation of order 2 i.e $\sigma_\Theta$ as the activation function).

| Architecture | Methods | Standard Accuracy | Provable Robust Acc. ($\rho =$) | | | Increase |
| --- | --- | --- | --- | --- | --- | --- |
| | | | 36/255 | 72/255 | 108/255 | (Standard) |
| LipConvnet-5 | SOC + $\mathrm{MaxMin}$ | 42.71% | 27.86% | 17.45% | 9.99% | _ |
| | **+ LLN** | 45.86% | 31.93% | 21.17% | 13.21% | +3.15% |
| | **+ HH** | **46.36%** | 32.64% | 21.19% | 13.12% | **+3.65%** |
| | **+ CR** | 45.82% | **32.99%** | **22.48%** | **14.79%** | +3.11% |
| LipConvnet-10 | SOC + $\mathrm{MaxMin}$ | 43.72% | 29.39% | 18.56% | 11.16% | _ |
| | **+ LLN** | 46.88% | 33.32% | 22.08% | 13.87% | +3.16% |
| | **+ HH** | **47.96%** | 34.30% | 22.35% | 14.48% | **+4.24%** |
| | **+ CR** | 47.07% | **34.53%** | **23.50%** | **15.66%** | +3.35% |

Table 11: Results for provable robustness against adversarial examples on the CIFAR-100 dataset.

| Architecture | Running times (in seconds) | | |
| --- | --- | --- | --- |
| | **MaxMin (no LLN)** | **MaxMin (LLN)** | **HH (LLN)** |
| LipConvnet-5 | 3.7568 | 3.5002 | 3.6673 |
| LipConvnet-10 | 5.3714 | 5.5482 | 5.5533 |
| LipConvnet-15 | 7.3092 | 7.2595 | 7.3127 |
| LipConvnet-20 | 9.005 | 9.2043 | 9.308 |
| LipConvnet-25 | 10.9321 | 10.7868 | 11.726 |
| LipConvnet-30 | 12.3198 | 13.2168 | 13.6275 |
| LipConvnet-35 | 14.427 | 14.575 | 15.7069 |
| LipConvnet-40 | 16.0911 | 16.2535 | 17.1015 |

Table 12: Inference times for various networks on the CIFAR-100 test dataset. Similar to CIFAR-10 (in Table 6), these numbers are for the whole test dataset with $10000$ samples.

| Architecture | Methods | Standard Accuracy | Provable Robust Acc. ($\rho =$) | | | Increase |
| --- | --- | --- | --- | --- | --- | --- |
| | | | 36/255 | 72/255 | 108/255 | (108/255) |
| LipConvnet-5 | BCOP | 46.31% | 31.55% | 20.34% | 12.52% | -0.69% |
| | Cayley | 44.61% | 31.01% | 19.84% | 12.43% | -0.78% |
| | SOC (baseline) | 45.86% | 31.93% | 21.17% | 13.21% | (+0%) |
| | **SOC + HH** | **46.36%** | 32.64% | 21.19% | 13.12% | -0.09% |
| | **+ CR** | 45.82% | **32.99%** | **22.48%** | **14.79%** | **+1.58%** |
| LipConvnet-10 | BCOP | 45.36% | 31.71% | 20.48% | 12.40% | -1.47% |
| | Cayley | 45.79% | 31.91% | 20.69% | 12.78% | -1.09% |
| | SOC (baseline) | 46.88% | 33.32% | 22.08% | 13.87% | (+0%) |
| | **SOC + HH** | **47.96%** | 34.30% | 22.35% | 14.48% | +0.61% |
| | **+ CR** | 47.07% | **34.53%** | **23.50%** | **15.66%** | **+1.79%** |
| LipConvnet-15 | BCOP | 43.70% | 30.11% | 19.85% | 12.29% | -1.47% |
| | Cayley | 45.05% | 31.60% | 20.32% | 12.93% | -0.83% |
| | SOC (baseline) | 47.72% | 33.52% | 21.89% | 13.76% | (+0%) |
| | **SOC + HH** | **47.72%** | 33.97% | 22.45% | 13.81% | +0.05% |
| | **+ CR** | 47.61% | **34.54%** | **23.16%** | **15.09%** | **+1.33%** |
| LipConvnet-20 | BCOP | 39.77% | 27.20% | 17.44% | 10.49% | -3.61% |
| | Cayley | 39.68% | 26.93% | 17.06% | 10.48% | -3.62% |
| | SOC (baseline) | 46.86% | 33.48% | 22.14% | 14.10% | (+0%) |
| | **SOC + HH** | 47.71% | 34.22% | 22.93% | 14.57% | +0.47% |
| | **+ CR** | **47.84%** | **34.77%** | **23.70%** | **15.84%** | **+1.74%** |

Table 13: Results for provable robustness on the CIFAR-100 dataset using shallow networks. All of these results include Last Layer Normalization (LLN).

| Architecture | Methods | Standard Accuracy | Provable Robust Acc. ($\rho =$) | | | Increase (108/255) |
| --- | --- | --- | --- | --- | --- | --- |
| | | | 36/255 | 72/255 | 108/255 | |
| LipConvnet-25 | BCOP | 34.15% | 21.57% | 13.52% | 7.97% | -5.89% |
| | Cayley | 33.93% | 21.93% | 13.68% | 8.19% | -5.67% |
| | SOC (baseline) | 46.32% | 32.87% | 21.53% | 13.86% | (+0%) |
| | **SOC + HH** | **47.70%** | 34.00% | 22.67% | 14.57% | +0.71% |
| | **+ CR** | 46.87% | **34.09%** | **23.41%** | **15.61%** | **+1.75%** |
| LipConvnet-30 | BCOP | 29.73% | 18.69% | 10.80% | 6% | -7.52% |
| | Cayley | 28.67% | 18.05% | 10.43% | 6.09% | -7.43% |
| | SOC (baseline) | 46.18% | 32.82% | 21.52% | 13.52% | (+0%) |
| | **SOC + HH** | 46.80% | 33.72% | 22.70% | 14.31% | +0.79% |
| | **+ CR** | **46.92%** | **34.17%** | **23.21%** | **15.84%** | **+2.32%** |
| LipConvnet-35 | BCOP | 25.65% | 14.88% | 8.47% | 4.30% | -8.95% |
| | Cayley | 27.75% | 16.37% | 9.52% | 5.40% | -7.85% |
| | SOC (baseline) | 45.22% | 32.10% | 21.28% | 13.25% | (+0%) |
| | **SOC + HH** | 46.21% | 32.80% | 21.55% | 14.13% | +0.88% |
| | **+ CR** | **46.88%** | **33.64%** | **23.34%** | **15.73%** | **+2.48%** |
| LipConvnet-40 | BCOP | 30.66% | 18.68% | 10.46% | 5.92% | -6.07% |
| | Cayley | 25.54% | 14.91% | 8.37% | 4.40% | -7.59% |
| | SOC (baseline) | 42.94% | 29.71% | 19.30% | 11.99% | (+0%) |
| | **SOC + HH** | **45.84%** | **32.79%** | 21.98% | 14.07% | +2.08% |
| | **+ CR** | 45.03% | 32.57% | **22.37%** | **14.76%** | **+2.77%** |

Table 14: Results for provable robustness on the CIFAR-100 dataset using deeper networks. All of these results include Last Layer Normalization (LLN).

| Architecture | Methods | Standard Accuracy | Provable Robust Acc. ($\rho =$) | | | Increase |
|---|---|---|---|---|---|---|
| | | | 36/255 | 72/255 | 108/255 | (108/255) |
| LipConvnet-5 | SOC + HH$^{(2)}$ | 46.61% | 32.50% | 21.34% | 13.22% | +0.01% |
| | **+ CR** | **46.69%** | **33.22%** | **22.34%** | **14.30%** | **+1.09%** |
| LipConvnet-10 | SOC + HH$^{(2)}$ | 47.47% | 33.32% | 21.84% | 13.75% | -0.01% |
| | **+ CR** | **47.53%** | **34.52%** | **23.06%** | **15.07%** | **+1.31%** |
| LipConvnet-15 | SOC + HH$^{(2)}$ | 47.19% | 33.67% | 22.36% | 13.78% | -0.09% |
| | **+ CR** | **47.22%** | **34.04%** | **22.98%** | **15.28%** | **+1.41%** |
| LipConvnet-20 | SOC + HH$^{(2)}$ | **47.86%** | 33.93% | 22.44% | 14.41% | +0.31% |
| | **+ CR** | 47.54% | **34.32%** | **23.53%** | **15.54%** | **+1.44%** |
| LipConvnet-25 | SOC + HH$^{(2)}$ | **47.86%** | 33.97% | 22.78% | 14.59% | +0.73% |
| | **+ CR** | 47.50% | **34.38%** | **23.92%** | **15.92%** | **+2.06%** |
| LipConvnet-30 | SOC + HH$^{(2)}$ | 46.23% | 32.64% | 21.95% | 14.00% | +0.48% |
| | **+ CR** | **46.36%** | **33.20%** | **22.70%** | **14.85%** | **+1.33%** |
| LipConvnet-35 | SOC + HH$^{(2)}$ | **46.06%** | 32.35% | 21.33% | 13.65% | +0.40% |
| | **+ CR** | 45.78% | **33.24%** | **22.39%** | **14.78%** | **+1.53%** |
| LipConvnet-40 | SOC + HH$^{(2)}$ | 43.81% | 30.59% | 20.08% | 12.56% | +0.57% |
| | **+ CR** | **45.61%** | **32.50%** | **22.36%** | **14.84%** | **+2.85%** |

Table 15: Results for provable robustness on CIFAR-100 using HH activation of Order 2 ($\sigma_\Theta$). Increase (108/255) is calculated with respect to SOC baseline (from Tables 13, 14). All of these results include Last Layer Normalization (LLN).

| Architecture | Methods | Standard Accuracy | Provable Robust Acc. ($\rho =$) | | | Increase |
|---|---|---|---|---|---|---|
| | | | (36/255) | (72/255) | (108/255) | (Standard) |
| LipConvnet-5 | SOC (no LLN) | 42.71% | 27.86% | 17.45% | 9.99% | (+0%) |
| | **SOC + LLN** | **45.86%** | **31.93%** | **21.17%** | **13.21%** | **+3.15%** |
| LipConvnet-10 | SOC (no LLN) | 43.72% | 29.39% | 18.56% | 11.16% | (+0%) |
| | **SOC + LLN** | **46.88%** | **33.32%** | **22.08%** | **13.87%** | **+3.16%** |
| LipConvnet-15 | SOC (no LLN) | 42.92% | 28.81% | 17.93% | 10.73% | (+0%) |
| | **SOC + LLN** | **47.72%** | **33.52%** | **21.89%** | **13.76%** | **+4.80%** |
| LipConvnet-20 | SOC (no LLN) | 43.06% | 29.34% | 18.66% | 11.20% | (+0%) |
| | **SOC + LLN** | **46.86%** | **33.48%** | **22.14%** | **14.10%** | **+3.80%** |
| LipConvnet-25 | SOC (no LLN) | 43.37% | 28.59% | 18.18% | 10.85% | (+0%) |
| | **SOC + LLN** | **46.32%** | **32.87%** | **21.53%** | **13.86%** | **+2.95%** |
| LipConvnet-30 | SOC (no LLN) | 42.87% | 28.74% | 18.47% | 11.21% | (+0%) |
| | **SOC + LLN** | **46.18%** | **32.82%** | **21.52%** | **13.52%** | **+3.31%** |
| LipConvnet-35 | SOC (no LLN) | 42.42% | 28.34% | 18.10% | 10.96% | (+0%) |
| | **SOC + LLN** | **45.22%** | **32.10%** | **21.28%** | **13.25%** | **+2.80%** |
| LipConvnet-40 | SOC (no LLN) | 41.84% | 28.00% | 17.40% | 10.28% | (+0%) |
| | **SOC + LLN** | **42.94%** | **29.71%** | **19.30%** | **11.99%** | **+1.10%** |

Table 16: Results for provable robustness on the CIFAR-100 dataset with and without LLN

