# OpenReview forum: "Improved deterministic l2 robustness on CIFAR-10 and CIFAR-100"
_ICLR.cc/2022/Conference — ICLR 2022 Spotlight_

### Official Review · Reviewer_dTST · 2021-10-24

**Correctness:** 4
**Technical Novelty And Significance:** 3
**Empirical Novelty And Significance:** 2
**Recommendation:** 8
**Confidence:** 2

**Main Review:**

This paper is well-written. I am not very familiar with the field of certified robustness, but the three bullet points listed in my Summary of Paper are well motivated and justified, and their success has been proved in numerical experiments. My major concerns are about the explanation of the numerical results. The authors have observed improvement in standard accuracy in certain circumstances, which needs more explanation on why this happened. Is it because you relax the orthogonalization requirement of the last linear layer?

A minor comment on Page 2, when introducing \rho, it would be good to define what it is. For readers who are not familiar with the field, it looks a bit confusing. I only got what it means when reading later sections of the paper.

Another comment at the end of Section 2, where you mentioned some past work focusing on l1 norm certificate. Can you expand a little bit explaining what is the motivation for considering l2 certificate, difference and applicable settings for these two concepts?

Lastly about the numerical section, is it possible to include results on large datasets such as ImageNet, to test the effectiveness, scalibility of the proposed procedure? Also, you observed different performance gains on different architectures. Can you provide an explanation on why this is so, and give a short summary regarding when your procedure would work best, and when it yields negligible gains?

**Summary Of The Paper:**

This paper proposed a procedure to certify the robustness of 1-Lipschitz CNNs by

(i) relaxing the orthogonalization of the last linear layer of the network, while requiring that all rows of the last layer matrix are normalized -- Last Layer Normalization;

(ii) introducing a certificate regularization that increases the robust certificate for correctly classified examples;

(iii) introducing a class of Gradient Norm Preserving activation functions called Householder activations which are necessary for any GNP piecewise linear function to be continuous.

They proved the success of the proposed procedure on CIFAR-10 and CIFAR-100 by showing improvements in provable robust accuracy with a comparable standard accuracy.

**Summary Of The Review:**

Overall I like this paper and the proposed ideas were well justified. My comments on the weakness mainly involve the explanation of the numerical results, and whether it is possible to include more results on larger datasets. It would be good if the authors can provide a short summary on when their method would work the best.

---

> ### Author Response · Authors · 2021-11-18
> **Author responses**
>
> **Explanation of improved results:** In Section 4 (LLN) and Section 5 (CR) of the paper, we discuss the reasons for the improved performance. We repeat the explanations below:
>
> ***LLN:*** We believe that LLN gives good performance because it relaxes the orthogonalization of the last layer. The original robustness certificate (used in the previous literature) is valid *only* when all layers in the network are orthogonal. Consider the matrix in the last linear layer in the network. Due to the orthogonality constraint, all rows of the matrix must always be orthogonal to each other during training, hence also be of unit norm. Any update made to some row of the matrix must require a change of direction (since it is unit norm) and that in turn must require a change in the direction of all other rows. This necessitates the forgetting of information already learned by the network. We have included a discussion about this phenomenon in Section 4.
>
> ***CR:*** The cross entropy loss for training is not explicitly designed to maximize the margin $\mathcal{M}_f(\mathbf{x})$ and thus, the robustness certificate. That is, once the cross entropy loss becomes small, gradients will no longer try to further increase the margin $\mathcal{M}_f(\mathbf{x})$ even though the network may have the capacity to learn bigger margins. Our introduced Certificate Regularization, regularizes the network to increase margins and hence the robustness certificate even after the cross entropy loss has become small.
>
> **Definition of** $\rho$ **:** We have updated the paper to include the definition of $\rho$ in the introduction on Page 2.
>
> **Differences between** $l_1$ **vs** $l_2$ **certificate:** We have added a discussion about the differences between $l_1$ and $l_2$ certificates in Appendix Section C. Here we briefly discuss it. Using the equivalence of norms, it can be shown that for $d$ dimensional inputs, a certificate of $\rho$ in the $l_1$ norm only implies a certificate of $\rho/\sqrt{d}$ in the $l_2$ norm (we prove the same in Appendix Section C). For CIFAR-10 and CIFAR-100, note that d = 32 x 32 x 3.  Thus, if we use the $l_1$ radius of $4$ used by Levine et al[1], we only get the $l_2$ certificate of $4/(32 \sqrt{3}) = 0.07225$ for CIFAR-10 and CIFAR-100. However, we certify robustness using $l_{2}$ certificates of much larger radii i.e $36/255 (0.14118)$, $72/255 (0.28236)$ and $108/255 (0.42354)$. Moreover, they only achieve 36\% certified accuracy at the $l_1$ radius of $4$ (which is the $l_2$ radius of $0.07225$) while we achieve more than 60\% certified robust accuracy using $l_2$ radius of $0.14118$.
>
> **Results on Imagenet:** It is difficult to scale 1-Lipschitz CNNs to Imagenet currently because of two reasons: (i) various orthogonal convolution layers run into memory overflow issues at that scale and (ii) difficulty of obtaining good results in 1-Lipschitz CNNs when the number of classes is large. We show significant improvements over prior works for CIFAR-100 in Table 1 and hence we believe this paper makes significant progress on the second problem. But obtaining strong results at the scale of Imagenet remains an open direction for future research.
>
> **Differences in performance across architectures:** We briefly discuss the performance improvements achieved by our introduced procedures below:
>
> ***Performance improvement due to LLN:*** Since we observe significantly improved results for CIFAR-100 and only negligible improvements for CIFAR-10, we believe that LLN is most effective in the regime when the number of classes is large.
>
> ***Performance improvement due to CR:*** We observe significant improvements in provably robust accuracy with only a small reduction in standard accuracy using Certificate Regularization across all architectures.
>
> ***Performance improvement due to HH:*** Based on our experiments, we observe that HH is most effective when the number of layers in the network is large.
>
> Overall, we believe our procedures work best either when the number of classes is large, or for deeper networks.
>
> [1] Improved, Deterministic Smoothing for $l_1$ certified robustness. Alexander Levine, Soheil Feizi. ICML, 2021.

---

### Official Review · Reviewer_zg69 · 2021-10-28

**Correctness:** 4
**Technical Novelty And Significance:** 3
**Empirical Novelty And Significance:** 3
**Recommendation:** 6
**Confidence:** 5

**Main Review:**

The main issue of this paper is evaluation. And the SOC work (Singla & Feizi, 2021) has the same issue.
The Lipschitz constant of an SOC layer is not strictly bounded by 1 and is only approximately 1.
Let X be the product of the Lipschitz constants of all layers in a network.
With many layers and multiplications, X could deviate sizably above 1.
This effect can get exacerbated after the extra loss term "Certificate Regularizer" (CR) is added which encourages the SOC layers to amplify logits and hence signals in all intermediate layers.
This explains why, in Table 2, the effect of CR grows stronger as the network gets deeper.

Therefore, the reported provable robustness numbers, which are measured by margins in logits, are not reliable:
* The absolute numbers of provable robust accuracies could be overestimated for all models in Tables 1 and 2.
* The relatively comparisons in Tables 1 and 2 are not with constant X. Since the comparisons are in the range of a few percentage points, the actual robustness difference between models can easily get lost in the fluctuation of X of different networks.

In such situation, please consider the following:
* Report measured robustness by running adversarial attacks. While doing that, the comparison should not be limited to SOC and should include other techniques like adversarial training.
* After training is done, calculate the actual X of the final network and then divide logits by X before measuring margins for provable robustness assessment.


**Summary Of The Paper:**

This paper addresses the problem of provable L2 robustness in image classification.
Three techniques are proposed as additions on top of the SOC work (Singla & Feizi, 2021):
1) let the last linear layer be non-orthogonal and adjust provable L2 robustness calculation accordingly;
2) add a loss term that encourages larger margins in logits;
3) a new nonlinearity.

Comparisons against the original SOC work are reported on CIFAR-10 and CIFAR-100.

**Summary Of The Review:**

The evaluation setup is flawed and therefore unable to judge the value of the proposed techniques. Update: ratings adjusted based on new results from authors.

---

> ### Author Response · Authors · 2021-11-18
> **Author responses**
>
> **Approximation error of an orthogonal matrix using SOC:** We respectfully disagree with this comment. To the best of our knowledge, SOC [1] is the only orthogonal convolution layer in the published literature that provides a provable guarantee on the approximation error to a true orthogonal matrix. Moreover, in [1], authors show that the approximation error is very small: they achieved a maximum bound on the approximation error of $2.42 \times 10^{-6}$ across all convolution layers in the network. Even if one uses a 40 layer network (the maximum number of convolution layers in our paper), the maximum possible Lipschitz constant of the network is $(1 + 2.415 \times 10^{-6})^{40} = 1.000097 \leq  1.0001$ which is essentially equal to 1 for all practical purposes. We use the exact same setup as used in SOC and thus obtain a similar bound on the approximation error. Even using Certificate Regularization for the deep network with 40 layers, we achieve the same bound on the approximation error. We have added a discussion about the same to the paper in Section 7 (second paragraph). Moreover, the accuracy improvements achieved by our work are of the order of 3-4 % in the standard and provably robust accuracy across different network architectures and datasets. A difference of 0.0001 is so small that it will hardly change any of the improvements achieved by our proposed methods.
>
> Please note that unlike SOC [1], other orthogonal convolution layers such as BCOP [2] and Cayley [3] do not provide any theoretical guarantees on the approximation errors. For example, the symmetric projector matrices used in BCOP [2] are also constructed using an orthogonal matrix. Any numerical error in the orthogonal matrix can lead to inexact symmetric projector which in turn can lead to the Lipschitz constant of the matrix deviating from 1. Similarly, Cayley convolution [3] uses the torch.inverse function to compute the inverse of the matrix. Any error in the computation of this inverse could also lead to a deviation of the lipschitz constant of the orthogonal convolution layer from 1. But Cayley and BCOP convolutions do not provide any theoretical bounds on these approximation errors while SOC does.
>
> [1] Skew Orthogonal Convolutions. Sahil Singla, Soheil Feizi. ICML, 2021.
>
> [2] Preventing Gradient Attenuation in Lipschitz Constrained Convolutional Networks. Qiyang Li, Saminul Haque, Cem Anil, James Lucas, Roger Grosse, Jörn-Henrik Jacobsen. NeurIPS, 2019.
>
> [3] Orthogonalizing Convolutional Layers with the Cayley Transform. Asher Trockman, J Zico Kolter. ICLR, 2020.

---

> > ### Comment · Reviewer_zg69 · 2021-11-19
> > **the best way to resolve the issue is measurement**
> >
> > Thanks for the explanation.
> >
> > However, the 2.42E-6 number from [1] was merely an example if the number of terms is 12 and if the Lipschitz of the skew-symmetric matrix is 1.8. It's unclear how that 1.8 number can be ensured. Spectral normalization is used, but that is an approximation itself because only one or a few power iterations are used per batch during training.
> >
> > The best way is to measure X as suggested in my review, and then divide logits by X before measuring margins for provable robustness assessment. The training is already done for the models in Tables 1 and 2, and it's now affordable to run many power iterations on the linear layers and calculate the actual X.
> >
> > This way Tables 1 and 2 would become a rigorous comparison and we'll see the real impact of the proposed techniques. It'd be even better if BCOP, Cayley and other competitors are added too and measured by the same approach.
> >
> > p.s. For the measurement, please do power iterations over a linear layer and not over a skew-symmetric matrix. Because it's more direct and reliable, and also it works on any pre-trained model and hence is apple-to-apple with different competitors.

---

> > > ### Author Response · Authors · 2021-11-19
> > > **Results on re-running the evaluations according to your proposed method**
> > >
> > > We use the publicly available implementation of SOC which performs spectral normalization using 50 power iterations after every 200 training steps. Additionally, we also evaluate using 50 power iterations during evaluation. This procedure ensures that the norm of the skew symmetric filter is theoretically bounded below 2.1. In our experiments we *empirically* observe the norm of skew symmetric filter to be bounded below 1.8 (i.e. slightly below the theoretical bound of 2.1).
> > >
> > > Nevertheless, to address your concerns, we conducted another evaluation for all the networks in Tables 1 and 2 using the exact setup proposed by you. That is, we computed the Lipschitz constant using power iteration on the linear layer *(without using the bound provided in the SOC paper for the skew symmetric filter).* Again, we used 50 power iterations to compute the Lipschitz constant per convolution layer. The revised numbers are given in Tables 4 and 5 in the Appendix Section D. For only 4 provable robust accuracy values in Table 1 (CIFAR-100) and 5 values in Table 2 (CIFAR-10), we observe a reduction of only 0.01%. **All the other numbers in the table remain completely unchanged (i.e. no change in provable robust accuracy values).**
> > >
> > > The cases for which the change of 0.01\% was observed are repeated in the Tables below. “$l_{2}$ radius” denotes the  $l_{2}$ radius at which the given provable robust accuracy numbers are computed, “Before” denotes the original provable robust accuracy at the specified $l_{2}$ radius and “After” denotes the provable robust accuracy at the same $l_{2}$ radius with the revised Lipschitz constant. As you can see, this change makes hardly any difference in the results demonstrated in the paper.
> > >
> > > **Results on CIFAR-100**
> > >
> > > | Network      | Methods | $l_2$ radius | Before | After |
> > > | ----------- | ---------------------- |  ----------- | ----------- | ----------- |
> > > | LipConvnet-20 | SOC + LLN + HH + CR                  |   108/255 | 15.84\% | 15.83\% |
> > > | LipConvnet-25   | SOC + MaxMin       |   72/255 | 18.18\% | 18.17\% |
> > > | LipConvnet-30   | SOC + MaxMin                |   108/255 | 11.21\% | 11.20\% |
> > > | LipConvnet-30   | SOC + LLN + HH + CR                |  108/255 | 32.57\% | 32.56\% |
> > >
> > >
> > >
> > > **Results on CIFAR-10**
> > >
> > > | Network      | Methods | $l_2$ radius | Before | After |
> > > | ----------- | ---------------------- |  ----------- | ----------- | ----------- |
> > > | LipConvnet-15 | SOC + MaxMin                  |   72/255 | 44.28\% | 44.27\% |
> > > | LipConvnet-20   | SOC + HH + CR       |   72/255 | 48.69\% | 48.68\% |
> > > | LipConvnet-25   | SOC + MaxMin                |   72/255 | 43.55\% | 43.54\% |
> > > | LipConvnet-30   | SOC + MaxMin                |  108/255 | 28.05\% | 28.04\% |
> > > | LipConvnet-40   | SOC + HH       | 72/255 | 42.24\% | 42.23\% |
> > >
> > > **We hope this convinces you about the validity of our experimental results. If you have any additional questions or comments, please let us know. If you are satisfied with our response, we kindly ask you to revise your evaluation of our work.**

---

> > > > ### Comment · Reviewer_zg69 · 2021-11-19
> > > > **ratings adjusted**
> > > >
> > > > Ratings adjusted based on the new numbers.

---

### Official Review · Reviewer_FsEx · 2021-10-29

**Correctness:** 4
**Technical Novelty And Significance:** 3
**Empirical Novelty And Significance:** 4
**Recommendation:** 6
**Confidence:** 4

**Main Review:**

Pros:
- Excellent numerical improvements
- The paper is clearly written
- The contributions are mainly from the engineering point of view and this paper demonstrates the method works well, with coherent explanations.

Cons:
- I have a minor concern w.r.t. the structure of the paper, I think some of the “Theorem” could be simply called “Proposition”: please don’t get me wrong, I simply think it’s good to distinguish crucial mathematical results from trivial properties (here continuous functions).
Theorem 1, I think, should be phrased differently: it requires to introduce formally the robustness certificate first. Then, one find a property of this certificate which can be stated as a proposition. Again, I feel Theorem 2 is a too strong statement for this. It’s also strange to see that the Theorem 1 uses phrasing from the Definition 1 but is still above this latter.
- I think the choice of the parameter $\gamma$ should be fully detailed. In other words, that it has been cross-validated on a subset of the training set. Otherwise, a suspicious reader could believe the accuracy has been optimized on the test set.

**Summary Of The Paper:**

This work is following the line of (Anil et al 2018), its goal is to propose criteria that allows to train models wirth orthogonal weights and to certify they are. The authors generalize (Anil et al 2018) by considering a more expressive non-linearity which leads to unitary jacobians, with a softer criterium during supervision which allows to obtain better accuracies. Overall, I feel the paper is well motivated, and the results are convincing.

**Summary Of The Review:**

The scientific contribution of this paper is solid in my opinion: a problem is introduced, solved with a generalization of a previous framework. The explanations of the improvements are clear, and because the contribution is fully technical and the results are good, I think, except if I missed an element (such as a working with a similar result) that this paper would be a nice contribution to ICLR.

---

> ### Author Response · Authors · 2021-11-18
> **Author responses**
>
> 1. **Use of Proposition instead of Theorem:** Thank you for your comment. Indeed, we agree that for some trivial properties, use of the word theorem wasn’t the best choice. We have updated the paper to rename Theorem 1 to Proposition 1 and Theorem 2 to Corollary 1. We have also re-ordered the paper so that Definition 1 appears before Corollary 1.
>
> 2. **Choice of** $\gamma$ **:** Indeed $\gamma$ was validated using a subset of $5000$ samples of the training set. We have included a discussion about our methodology for selecting $\gamma$ in Appendix Section B.

---

### Official Review · Reviewer_imTR · 2021-11-03

**Correctness:** 3
**Technical Novelty And Significance:** 3
**Empirical Novelty And Significance:** 3
**Recommendation:** 8
**Confidence:** 4

**Main Review:**

Pros:
1. The authors find several limitations of existing works and propose three novel methods to address them. Both methods are very novel and have their advantages to address the problems.
2. The authors present their findings and current problems and introduce the proposed methods very clear from a high level to a very detailed level. The notations, equations, and proofs are easy to follow.
3. The figures and tables are clearly designed. Figure 1 illustrates the HH activation very clear and makes it easier to be understood. We can clearly see the improvements achieved by each method from Table1.
4. This paper contributes a lot to the research area of 1-Lipschitz CNNs, especially the class of piecewise linear GNP activation functions HH.

Cons:
1. A minor issue is the structure of this paper. In the Introduction, the authors present the three methods in the order: HH, LLN, CR. However, the order in the method sections is LLN, CR, HH. The logic is a little bit messed up.
2. It is not clearly explained why the original robustness certificate (Eq 4.) has a problem when the number of class k is large. And why LLB can not achieve significant gains on CIFAR10, just because of the small number of classes?

Question:
1. From Table 1, why the performance of LipConvnet-40 (45.84) is even worse than LipConvnet-5 (46.36) and other shallow networks?

**Summary Of The Paper:**

The authors propose three methods to improve deterministic l2 robustness certificates of 1-Lipschitz convolutional neural networks.

1. Last Layer Normalization (LLN):
    (1) Relax the orthogonalization of the last weight layer of the network W.
    (2) The direction of the rows in W can change freely; do not need to update other rows in W and forgot the learned information.
    (3) Efficient than original robustness certificate when the number of classes K is large.
2. Certificate Regularizer (CR):
    (1) Enable the gradients to continue increasing the margin M_f(x) while entropy loss becomes small by adding the regularization term.
    (2) The relu function in the maximization term ensures that the optimization tries to increase the certificates only for the correctly classified inputs.
3. Householder Activation Functions (HH):
    (1) Introduce a general linear GNP activation function named HH for the special case MaxMin used in Anil et al., 2018. and other works.
    (2) Prove this HH transformation is necessary for any GNP piecewise linear function to be continuous.

Their experimental results demonstrate the effectiveness of these proposed methods compared to the baseline method SOC.

**Summary Of The Review:**

The paper proposed three novel methods to address some limitations of existing works. It appears to be technically sound, but I have not carefully checked every detail. The experimental evaluations convincingly support the main claims. This paper contributes some new ideas in this specific research area.

---

> ### Author Response · Authors · 2021-11-18
> **Author responses**
>
> 1. **Order of methods:** In the updated version of the paper, we have changed the order of HH, LLN, CR to LLN, CR, HH in the introduction.
> 2. **Details about LLN:** The original robustness certificate in equation 3 (in the updated draft of our paper) does not give good results for large number of classes because it is valid *only* when all layers in the network are orthogonal. Consider the matrix in the last linear layer in the network. Due to the orthogonality constraint, all rows of the matrix must always be orthogonal to each other during training, hence also be of unit norm. Any update made to some row of the matrix must require a change of direction (since it is unit norm) and that in turn must require a change in the direction of all other rows. This necessitates the forgetting of information already learned by the network. We have included a discussion about this phenomenon in Section 4.
> Indeed, we believe that small improvement on CIFAR-10 using LLN is due to the smaller number of classes in CIFAR-10. We discuss the same in Section 7.2.
> 3. **LipConvnet-40 performance:** Residual connections and batch normalization layers are critical for training very deep convolutional neural networks such as Resnet-1001. However, for 1-Lipschitz CNNs, it is difficult to maintain the 1-Lipschitzness of the network when any of these layers are used. This limitation of 1-Lipschitz CNNs is discussed in more detail in Li et al [1].
> We believe this could be the factor responsible for the performance of LipConvnet-40 being worse than that of LipConvnet-5.
>
>
> [1] Preventing Gradient Attenuation in Lipschitz Constrained Convolutional Networks. Qiyang Li, Saminul Haque, Cem Anil, James Lucas, Roger Grosse, Jörn-Henrik Jacobsen. NeurIPS, 2019.

---

> > ### Comment · Reviewer_imTR · 2021-11-22
> > **Thank you for your clarification**
> >
> > The detailed responses are clear and have addressed my major concerns.  It is helpful to understand why LLN is more efficient than the original robustness certificate when the number of classes K is large. Great contributions.
> > The rating remains unchanged.

---

### Decision · Program_Chairs · 2022-01-20

**Decision:**

Accept (Spotlight)

**Comment:**

The paper provides a procedure for certifying L2 robustness in image classification. The paper shows that the technique indeed works in practice by demonstrating it's accuracy on CIFAR-10 and CIFAR-100 datasets.

The reviewers are positive about the paper. Please do incorporate feedback, especially around experimental setup to ensure that the work compares various methods fairly and provides a clear picture to the reader.